**Investigation**

# How group structure impacts the numbers at risk for coronary artery disease: polygenic risk scores and nongenetic risk factors in the UK Biobank cohort

Jinbo Zhao,[1,2] Adrian O'Hagan,[1,2] Michael Salter-Townshend [ID] [2,*]

[1]Insight Centre for Data Analytics, University College Dublin, Belfield, Dublin D04V1W8, Ireland
[2]School of Mathematics and Statistics, University College Dublin, Belfield, Dublin D04V1W8, Ireland

*Corresponding author: School of Mathematics and Statistics, University College Dublin, Belfield, Dublin D04V1W8, Ireland. Email: michael.salter-townshend@ucd.ie

The UK Biobank (UKB) is a large cohort study that recruited over 500,000 British participants aged 40–69 in 2006–2010 at 22 assessment centers from across the United Kingdom. Self-reported health outcomes and hospital admission data are 2 types of records that include participants' disease status. Coronary artery disease (CAD) is the most common cause of death in the UKB cohort. After distinguishing between prevalence and incidence CAD events for all UKB participants, we identified geographical variations in age-standardized rates of CAD between assessment centers. Significant distributional differences were found between the pooled cohort equation scores of UKB participants from England and Scotland using the Mann–Whitney test. Polygenic risk scores of UKB participants from England and Scotland and from different assessment centers differed significantly using permutation tests. Our aim was to discriminate between assessment centers with different disease rates by collecting data on disease-related risk factors. However, relying solely on individual-level predictions and averaging them to obtain group-level predictions proved ineffective, particularly due to the presence of correlated covariates resulting from participation bias. By using the Mundlak model, which estimates a random effects regression by including the group means of the independent variables in the model, we effectively addressed these issues. In addition, we designed a simulation experiment to demonstrate the functionality of the Mundlak model. Our findings have applications in public health funding and strategy, as our approach can be used to predict case rates in the future, as both population structure and lifestyle changes are uncertain.

Keywords: UK Biobank; coronary artery disease; polygenic risk score; pooled cohort equation risk; group structure

## Introduction

### Coronary artery disease

Coronary artery disease (CAD), sometimes referred to as coronary heart disease (CHD) or ischemic heart disease, is one of the leading causes of morbidity and mortality in the United Kingdom, the United States, and worldwide (e.g. Cheema *et al.* 2022; Shahjehan and Bhutta 2022). Many environmental factors including smoking, unhealthy diet, alcohol intake, obesity, hypertension, diabetes mellitus, and lack of physical activity, impact the development of CAD (Mack and Gopal 2016). Family history of cardiovascular disease (CVD) has been extensively researched as a standalone risk factor for CAD both in the short and long term (e.g. Lloyd-Jones *et al.* 2004; Bachmann *et al.* 2012).

Several risk scores have been proposed to estimate the future cardiovascular risk (e.g. over the next 10 years) for currently healthy people, such as the Framingham risk score (FRS) (D'Agostino *et al.* 2008), QRISK3 (risk score using the QRESEARCH database) (Hippisley-Cox *et al.* 2017), and pooled cohort equation (PCE) scores (Goff *et al.* 2014). These scores combine the effects of multiple carefully selected nongenetic risk factors into a single score, and the effect of each risk factor or interaction term is estimated through sophisticated statistical analysis. Family history is included in QRISK3, but not in the other 2 scores. Those

overall risk scores are clinically relevant; for example, if a currently healthy person is diagnosed with a PCE-estimated 10-year CVD risk exceeding 7.5%, they will be advised to take statin therapy to reduce their future cardiovascular risk after consultation with their doctor in the United States (Vasan and Van den Heuvel 2022).

Our understanding of the genetic structure of CAD is also increasing. Genotyping microarrays designed to capture most common interindividual genetic variation provide the basis for genome-wide association studies (GWAS) (Khera and Kathiresan 2017). Since the first GWAS on CAD reported 3 common variants associated with increased risk of CAD, more than 250 putatively causal variants have been identified (Aragam *et al.* 2022). Causal variants are those that have biological effects on the phenotypes (Hormozdiari *et al.* 2015), but GWAS also detects many genetic variants that have no biological effect but are statistically significantly associated with phenotypes (Visscher *et al.* 2017).

Efforts to predict human diseases and traits by leveraging GWAS include polygenic risk scores (PRS). Essentially an individual's risk is estimated as a weighted sum of their personal SNPs, with weights equal to the GWAS based estimated effects. See CAD-PRS set selection for details. These have the potential to be useful in clinical settings, particularly in the context of specific purposes and conditions (Ogbunugafor and Edge 2022). PRS is

a tool that translates personal genetic information into real numbers that can be interpreted as an individual's genetic risk for a particular disease. There is already compelling evidence indicating its effectiveness in predicting the risk of CAD at the individual level (Dikilitas et al. 2022).

The combination of genetic and nongenetic risk factors increases the predictive power at the individual level. Elliott et al. (2020) calculated CAD-PRS and PCE scores for their study participants and compared the predictive power of risk factors alone and combined. They found that the overestimation of risk by PCE scores could be corrected by adding CAD-PRS to the model. Comparing the model with only PCE to the model with PCE and PRS, when using a risk threshold of 7.5%, the latter improved net reclassification 4.4% for cases and −0.4% for controls. Incorporating family history and PRS can improve the accuracy of predicting CAD risk in both real-world and simulation study settings (e.g. Hujoel et al. 2022; Zhao et al. 2023).

## Geographical variations in cardiovascular disease prevalence across the United Kingdom

CVD is the term for all types of diseases that affect the heart or blood vessels and CAD is the most common type of CVD. Within the United Kingdom, the higher prevalence of CVD in Scotland than in England has been repeatedly observed (e.g. Lawlor et al. 2003; Bhatnagar et al. 2016). The recent epidemiology study conducted by Cheema et al. (2022) shows the age-standardized CVD mortality rate differences in 2019 across 13 UK regions/nations, including the East Midlands, East England, London, Yorkshire, the Humber, Wales, and Scotland. Among those regions, Scotland has the highest mortality rate per 100,000 for CVD for all ages.

Environmental and genetic risk factors can both contribute to geographical variations in CVD (e.g. Lawlor et al. 2003; Peasey et al. 2006; Ding and Kullo 2009). For example, Lawlor et al. (2003) concluded that age distribution, socioeconomic status (SES), and health service utilization were the main causes of geographical variation, as well as differences in risk factors associated with CVD, including smoking, hypertension status, blood pressure, and cholesterol levels. Ethnic-specific differences in the genetic architecture of CAD have been widely proposed and explored, and different novel disease-susceptibility loci have been identified in different populations (Miyazawa and Ito 2021).

Geographical variations in CAD prevalence were reflected in the UK Biobank (UKB) participants, with CAD prevalence of 7.73% in England UKB participants and 9.06% in Scotland UKB participants (Yang et al. 2021). Yang et al. (2021) conducted a study on UKB participants to explore whether environmental or genetic factors could explain the regional CAD prevalence differences. They calculated the FRS, QRISK3, and PRS for CAD risk and concluded that neither FRS, QRISK3, nor PRS could explain the higher CAD prevalence in Scotland. They used Pearson's $\chi^2$ test and the 2-tailed Mann–Whitney test for statistical analysis. However, because they observed significant differences in the distribution of individual risk alleles, they concluded that the genetic architecture of a common disease could be different for geographically and ethnically closely related populations.

A well-known issue with polygenic scores is the lack of portability to populations that are genetically drifted from the samples used to train the model. It has been shown that accuracy declines with divergence from the training set across populations. Scutari et al. (2016) and Martin et al. (2017) emphasized the importance and challenges of developing generalized risk prediction methods for use across multiple populations. One commonly used technique to account for population stratification within a set of samples is to include some number of principal components (PCs) from a PCA of the genotypes in the regression equation used to construct the polygenic scores. However, Lin et al. (2023) (also using UKB data) showed that factors such as age, sex, genetic batch, and assessment center potentially exert a greater influence on PRS predictions compared to the inclusion of the PCs; indeed, Lin et al. (2023) claimed that "only up to three PCs appears to be sufficient for controlling population stratification for most outcomes, whereas including other covariates (particularly age and sex) appears to be more essential for model performance."

Our study showed that adding group-specific means of covariates greatly improves the estimation of group-specific rates, while also improving prediction at the individual level. Two recent papers investigated the causality and mechanism of population structure (or ancestry) on Polygenic Risk Score accuracy variation: Hou et al. (2023) and Hu et al. (2023). In short, both papers showed that the underlying causal effect sizes in GWAS were actually very similar (or even the same) in different ancestries. They showed this by examining admixed individuals and including (estimated) local ancestry in the GWAS model, leading to estimates of causal SNP effects that are highly consistent across different ancestries (e.g. European and African). However, they arrived at different conclusions: Hou et al. (2023) concluded that the differences observed across populations were due to gene–environment interactions, because they controlled for the environment and then got similar effect size estimates. In contrast, Hu et al. (2023) did not control for the environment, but did include local ancestry (specific to the chromosome segment) of each individual (Hou et al. 2023 included genome-wide averages only). They still obtained consistent estimates of effect sizes across populations, so they concluded that there was no cross-ancestry difference for either gene-gene or gene–environment interactions. They noted that gene–environment interactions did impact prediction accuracy, but acted in similar ways across different populations. In other words, the interactions worked the same, but differing environments led to different contributions to risk via gene–environment terms. Our proposed model leverages group-specific adjustments using covariates. The group-specific mean terms we proposed to use need not be directly or causally contributing to risk, we only require that there was some nonzero correlation with potentially unobserved contributions to risk for our model to benefit from an increased accuracy.

## Study aim

Genetic and nongenetic risk factors working together can improve the prediction of CAD risk at the individual level (e.g. Elliott et al. 2020; Hujoel et al. 2022), but few studies have used them jointly to estimate the number of risks at the regional/country level. Recently, Jain et al. (2023) reported 4 out of 14 tested common disorders as having a statistically significant correlation between observed population-wide prevalence and mean PRS of a sample of individuals from that population across Europe. They reported twice as many (8) of the disorders as having a statistically significant correlation across a global set of countries. They concluded that their "work can also be expanded to … identify populations who could be at a higher risk for severe symptoms due to specific environmental factors."

Eletti et al. (2022) described quantifying disparities of cancer survival between different subgroups (such as those defined by geography) as one of the primary aims of population cancer epidemiology. They stated that "Cancer research strives to provide an accurate picture of the evolving cancer burden, as well as documenting existing inequalities, using a variety of key indicators, including cancer survival." Similarly, in this study, we were

interested in comparing, analyzing and predicting the risk of CAD at a regional level, with a focus on characterizing the contribution of genetic and nongenetic effects to group-level rates. This will not only enhance our understanding of the mechanisms of CAD, but potentially allow for forecasting based on projected changes to environment and population genetic structure. Unlike Jain *et al.* (2023), we improved upon the use of group-mean risk estimates as the estimator, and our proposed model also sheds light on the interaction of genetic and nongenetic risk factors at a group level.

Study participants and regional selection are explained in section *Study participants and CAD events*, followed by the test methods used to compare the distribution of PCE and a different set of CAD-PRS at the group level. Section *Results for UKB assessment centers* contains predictions of the number of people at risk for CAD at the regional level using a generalized linear model (GLM) regressed on PCE and PRS, with a poor ability to distinguish between high and low case rate groups. The results in Ascertainment bias confounds group-rate estimation show that it is ascertainment bias that confounds group-rate estimation. Participation bias is common in population-based cohort studies, including the UKB study, and can bias the results of genetic epidemiology studies (Schoeler *et al.* 2023). The Mundlak GLM (Dieleman and Templin 2014) is used to eliminate bias and improve efficiency. A simulation experiment (section *How the Mundlak model works* and section *Structured permutations results*) is designed to explain how the Mundlak model works, in the presence of group structure that causes poor performance of standard regression models in predicting numbers at risk.

## Methods

### Study participants and CAD events

#### UKB resources and health outcomes records

The UKB study (Sudlow *et al.* 2015) recruited half million UK participants aged 40–69 from across the United Kingdom during 2006–2010 for the baseline assessments. Over 70% of all UKB participants are from England, <10% are from Scotland, and the rest are from Wales, Northern Ireland, and other regions. The baseline assessments were conducted at 22 assessment centers in Scotland, England, and Wales and consisted of a 5-part assessment process lasting 2–3 h. Figure 1 is the map from the UKB website (https://www.ukbiobank.ac.uk/enable-your-research/about-our-data/baseline-assessment) showing the locations of the UKB assessment centers. The process included written consent, answering touch screen questionnaires, face-to-face interviews with a study nurse, measurements like hand grip and bone density, and the sample collection of blood, urine, and saliva. The collected samples were used for gene sequencing and biochemical markers measurement, with various types of genetic data released since May 2015 (Bycroft *et al.* 2018).

UKB resources provide 2 types of record containing participants' disease status, self-reported health outcomes, and hospital inpatient data. Participants were asked to report their health outcomes during the baseline assessment, including the type of disease(s) and the date(s) of onset. Additionally, UKB also keeps track of each participant's hospital inpatient data, including hospital admissions information and date of admission, diagnosis during admission, procedures, and discharge information. For example, hospital inpatient data for UKB participants from England are provided by the Data Access Request Service (DARS), managed by National Health Service (NHS) digital, and provides hospital inpatient admissions data for English participants. Inpatient data

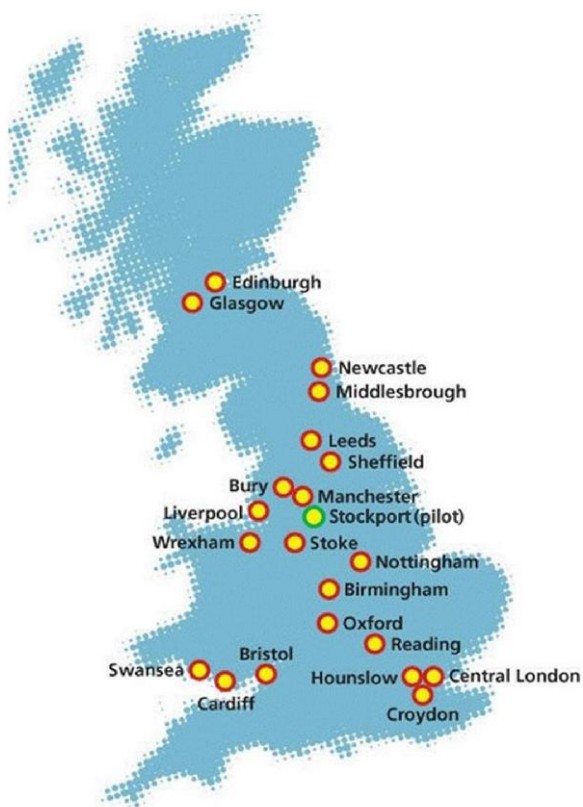

**Fig. 1.** Locations of the UKB baseline assessment centers, taken from https://www.ukbiobank.ac.uk/enable-your-research/about-our-data/baseline-assessment.

for participants from Wales and Scotland are provided via different partnerships. The UKB resources have over 10,000 data fields, with more arriving all the time. Those data fields can be assigned to several categories, such as physical measurements, lifestyle, cognition and hearing, physical activity, imaging, biomarkers, and genetics. Hospital diagnoses information accounts for almost half of all UKB data fields (Madakkatel *et al.* 2021). UKB participants' health outcomes are accessed by different coding systems for self-reported records and for the hospital inpatient records. Detailed, self-reported health outcomes are recorded separately for cancer and noncancer conditions using UKB designed datacoding. All clinical data in the hospital inpatient data are coded according to the World Health Organization's International Classification of Diseases (ICD) and all operations and procedures in the hospital inpatient data are coded according to the Office of Population, Censuses, and Surveys: Classification of Interventions and Procedures (OPSC) (UK Biobank: hospital inpatient data).

### CAD definition

To identify UKB participants diagnosed with CAD, CAD codes within self-reported and hospital inpatient records need to be determined first. There is no precise definition of which diseases should be included in determining the onset of CAD for UKB participants. Our study followed the CAD definition from Elliott *et al.* (2020). In detail, 6 different categories were searched to determine CAD events, including ICD-10, ICD-9, OPCS-4, noncancer illness code, operation code, and the vascular/heart problems data field. The CAD definition is in Supplementary Table S1 and the related UKB data fields are in Supplementary Table S2.

We defined any CAD events that happened before the date of joining the UKB for the initial assessment as prevalence CAD, and any events that happened after joining the UKB as incidence CAD. Some participants had more than 1 CAD events in their records, either 1 category with at least 2 different types of CAD events, or more than 2 categories of CAD events. For those cases, we compared the dates for multiple events and kept the earliest CAD event in this study. ICD-10, ICD-9, and OPCS-4 have the CAD date, while the other 3 have the CAD onset age in integer values. There may be some bias in converting the date of CAD onset to age at CAD onset to determine the first CAD event, as the date of birth of the UKB participants was not available in this study, only the year of birth.

### UKB assessment centers to represent geographical regions

After we identified the prevalence and incidence of CAD events, we calculated the age-standardized prevalence in 2010 and 2021 across the assessment centers. Of all 22 centers, only 2 were located in Scotland, the rest were in England and Wales, and one of the centers in England was a pilot center for only the first month of the overall baseline assessment period and had a relatively small number of participants. The UKB team sent invitation letters to people who were predominantly located in urban areas and lived near any of the UKB assessment centers (van Alten *et al.* 2022). Therefore, it is reasonable to use the assessment center as a geographic location to compare CAD morbidity. To obtain the age-standardized prevalence rates, we also used the 2013 European Standard Population as in Cheema *et al.* (2022). Note that Swansea and Wrexham are mobile assessment centers; as such they have far fewer samples associated with them and differing collection methods, leading to potential ascertainment bias.

## Genetic and nongenetic risk scores
### CAD-PRS set selection

The basis of PRS is that for most common diseases, their inheritance involves many common genetic variants with small effects, and combining those effects together has the ability to distinguish risk groups. The calculation process for PRS is complex and beyond the scope of this paper, but interested readers can learn more from Choi *et al.* (2020). The baseline function for PRS using additive genetic models summarizes the effects of a set of significant genetic variants, with the number of genetic variants varying from hundreds to several millions. Various PRS methods have been developed aimed at determining the set of variants included in the baseline calculation (e.g. Chang *et al.* 2015; Ge *et al.* 2019) and/or to estimate the magnitude of the effect (e.g. Vilhjálmsson *et al.* 2015; Mak *et al.* 2017).

The UKB resources category 300 provides access to standard PRS and enhanced PRS for 28 diseases (including CAD) and 25 quantitative traits, with the standard set (centered and variance-standardized) calculated for all participants in the UKB using algorithms trained on external data only (Thompson *et al.* 2022). They built their PRS algorithms using a Bayesian approach, combing data across multiple ancestries, using 9 different GWAS resources totalling 233,928 cases and 1,606,361 controls. They defined CVD as including any of narrow CAD, ischemic heart disease wide definition, ischemic stroke excluding all hemorrhages, broad CAD, major CHD event.

### Calculation of pooled cohort equation scores

The American College of Cardiology (ACC) and the American Heart Association (AHA) developed pooled cohort equations (PCE) to estimate the composite endpoint of 10-year atherosclerotic cardiovascular (ASCVD) risk, with initial sex-specific and ethnicity-specific equations published in 2013 (Goff *et al.* 2014). Atherosclerosis is a common disease that occurs when a sticky substance called plaque builds up inside your arteries. ASCVD events include CAD, stroke, and peripheral artery disease (PAD) (DeFronzo and Ferrannini 1991). The PCE tool is a risk assessment method that has been developed based on data that can be easily collected by primary care providers and can be implemented in routine clinical practice. Carefully selected risk factors associated with CAD risk are included in PCE equations, including age, total, and high-density lipoproteins (HDLs) cholesterol levels, blood pressure, smoking status, diabetes mellitus, and hypertension medication status. Log transformation and interaction terms are included in the equations. The PCE score is a single score that summarizes the effect using the parameters estimated by the proportional hazards model. The PCE scores for UKB participants have been studied widely, such as in Riveros-Mckay *et al.* (2021) and Carter *et al.* (2022).

There are criteria for applying the PCE equation. Stone *et al.* (2014) points that it is not appropriate to estimate 10-year ASCVD using PCE scores for individuals with clinical ASCVD, or with LDL-C ≥ 190 mg/dl, or people who are already in a statin benefit group.

We firstly identified UKB participants who already had an ASCVD event prior to joining the UKB, as the risk factors used to calculate PCE scores were collected at the baseline assessment visit. This study used the definitions of CVD from Elliott *et al.* (2020) to determine the prevalence CHD and stroke events, and the definition of PAD from Klarin *et al.* (2019), with relevant data fields from UKB, is in Supplementary Table S4. Following the CAD prevalence definition in section *Study participants and CAD events,* the prevalence ASCVD events were determined by comparing their event onset dates with the date they joined the UKB. The corresponding events codes are in Supplementary Table S2. Of the 502,401 UKB participants, 35,463 were identified as participants with a first ASCVD episodic event and were excluded because their ASCVD events occurred before they joined the UKB. Only 2 UKB participants had LDL-C ≥ 190 mg/dl during their initial assessment visit. Finally, we selected UKB participants who were already on statin therapy prior to joining UKB. We used the types of statin (atorvastatin, simvastatin, fluvastatin, pravastatin, and rosuvastatin) listed by Carter *et al.* (2022).

This study employed the PCE coding provided in the supplementary material of Vasan and Van den Heuvel (2022) to calculate PCE risk scores. We also followed their additional criteria that PCEs were not applied for people with extreme total cholesterol (>320 or <130 mg/dl), HDL cholesterol (>100 or <20 mg/dl), or systolic blood pressure (>200 or <90 mm Hg). The risk factors associated with the UKB data fields are listed in Supplementary Table S5.

### Study flow

The complete data set for this study included UKB participants who were eligible for PCE risk calculation and had CAD-PRS provided by UKB. In addition, body mass index (BMI) and Townsend deprivation index (TDI) were also extracted from the UKB resource for those participants, as BMI has been recognized as a risk factor that could aid the predictive power of PRS (e.g. van Alten *et al.* 2022) and TDI, a measure of social deprivation, has impact on the mortality of CVD (e.g. Ford and Highfield 2016). Figure 2 is the flow chart of obtaining this full data set for the following analysis. The complete data set had a total of 263,087 UKB

participants, with 8,458 participants developing CAD after enrolling in the UKB and the remaining 254,629 participants remaining CAD-free.

## Statistical tests

We examined the difference in PCE scores between England and Scotland, using a 2-tailed Mann–Whitney test as per Yang *et al.* (2021). The Mann–Whitney test is a nonparametric test and checks if 2 samples come from the same distribution by comparing the probability of X being greater than Y with the probability of Y being greater than X after randomly selecting values from sets X and Y (Hart 2001). We checked for but did not find evidence of pervasive statistically significant differences between assessment centers using Cohen's D. However, this is because those test the ratio of difference in mean to pooled standard deviation and as such we believe it is examining the wrong statistic. Two populations with the same mean PRS, but different variances, will have differing rates as the cases will arise predominantly from individuals in the upper tails of the distributions. We were able to record the "true" total liability of each individual then two populations with the same mean liability could have all cases arising from only the population with sufficiently high variance to exceed the threshold for disease.

We, therefore, employed a permutation test to determine whether 2 groups differ in the distribution of PRS or PCE values. In this study, we used the assessment center to represent the geographical region, but we noted that the number of people going to an assessment center close to their address was much lower than the number of people in that area. When we do not have access to the PRS of everyone in the region, but still want to compare the distributions of PRS, permutation tests are useful (Irizarry and Love 2016). People with PRS in the highest polygenic risk group have a higher chance of developing the disease than people with

average PRS scores. For example, Lewis and Green (2021) examined the ability of PRS to predict risk for CAD using genotype and phenotype data from UKB participants, as the highest polygenic risk group had twice the hazard ratio of the intermediate risk group. Therefore, when comparing the PRS distributions of 2 populations, we are more interested in looking at the tails or spread of the PRS distribution than just comparing the means or variances.

The permutation test is a resampling and nonparametric test that does not make any assumptions about the distribution. A full permutation test encompasses all possible permutations, here we use a large number of random permutations, hence it is a Monte Carlo permutation test. Our permutation test requires 4 main steps:

1) determine and calculate the statistic of interest (e.g. mean, median, or variance);
2) combine groups together, retaining all data but randomly shuffling the groups' labels, and then calculate the new statistic values;
3) repeat step 2 2,000 times and keep a record of the new statistic values;
4) the P-value is the proportion of statistics from the real group lower than the statistics from the reshuffled groups;
5) the Benjamini & Hochberg (BH) correction (Benjamini and Hochberg 1995) is applied to the P-values obtained in step 4 to control the false discovery rate as multiple hypothesis tests are performed in order to compare all pairs of assessment centers.

We also performed permutation tests on the predicted disease risk using the liability threshold model (LTM). The LTM assumes that there is a hidden continuous disease liability $L$ that

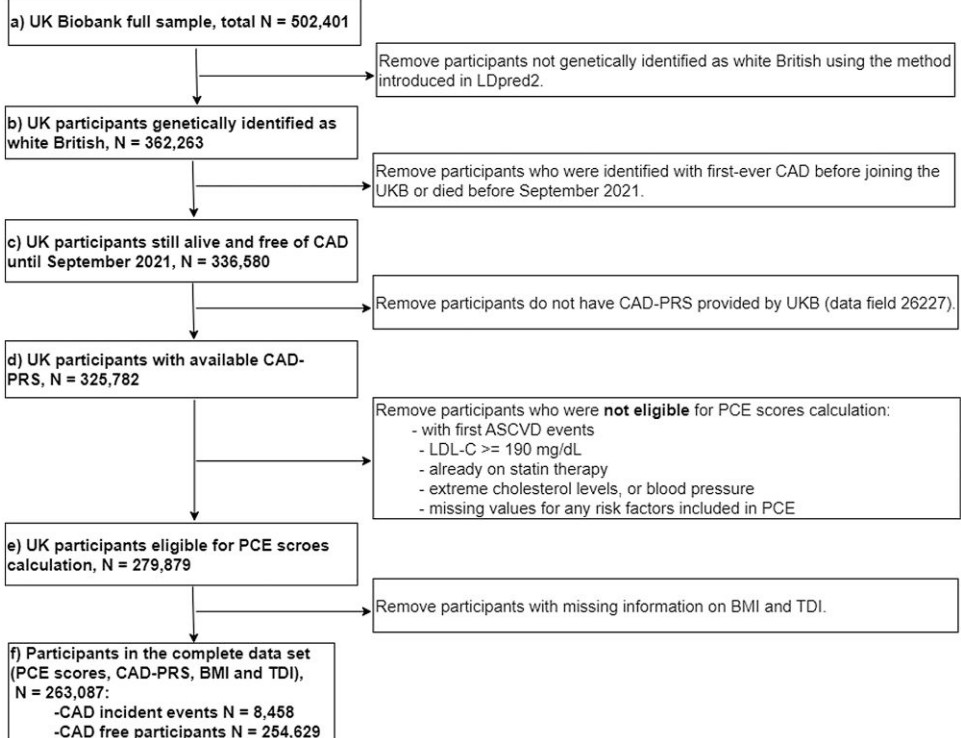

**Fig. 2.** Study flow chart to generate the complete data set for further analysis. PCE, pooled cohort equation; PRS, polygenic risk score; BMI, body mass index; TDI, Townsend deprivation index; ASCVD, atherosclerotic cardiovascular; LDL-C, low-density lipoprotein cholesterol.

determines the binary disease outcome, where $L$ follows a standard normal distribution, and the binary outcome $D = 1$ if $L$ exceeds a fixed threshold $T$ and 0 otherwise. The threshold is determined by the prevalence $K$ of the disease in the population using the relationship $T = \Phi^{-1}(1 - K)$, where $\Phi$ is the cumulative distribution function of the normal distribution (So *et al.* 2011). The total liability $L$ is assumed to be split into 2 components, the measurable genetic component and the combination of environmental and unknown risk factors, while PRS can be used to represent the measurable genetic component (Zhao *et al.* 2023). If we assume that the variance explained by PRS is $V$, then the LTM suggests that $\text{Cov}(L, \text{PRS}) = \text{Var}(\text{PRS}) = V$ and $\mathbb{E}[L \mid \text{PRS} = \text{prs}_j] = \text{prs}_j$ and $\text{Var}(L \mid \text{PRS} = \text{prs}_j) = 1 - V$. Then, using standard regression theory, we can calculate the probability of individual $j$ being a case given their value of PRS as $\Pr(L > T \mid \text{PRS} = \text{prs}_j) = \Pr(L - \text{prs}_j > T - \text{prs}_j) = 1 - \Phi(T - \text{prs}_j, 0, 1 - V)$ (So *et al.* 2011).

In this study, we denoted the CAD incidence rates from the complete data as $P$, and calculated the predicted probability of being a case after standardizing the CAD-PRS. The reason for this approach is that LTM provides a framework for predicting the risk of developing the disease solely based on PRS values within a specific group. Thus, 2 cohorts (whether observed/test center or randomly permuted groups) may have identical mean or median PRS values, but the proportion of individuals exceeding the threshold PRS may differ.

## Generalized linear model to predict the number of risk

A GLM regression is used to predict the probability of developing CAD for every sample in the complete data set, as detailed in section *Study flow*. When regressed on PRS and PCE, the model is

$$\text{logit}(p_j) = \beta_0 + \beta_{\text{PRS}}\text{PRS}_j + \beta_{\text{PCE}}\text{PCE}_j, \quad (1)$$

where $j \in (1, J)$ denotes the observation from the complete data set, $\beta_0$ is the intercept, $\beta_{\text{PRS}}$ and $\beta_{\text{PCE}}$ are the regression coefficients for the respective variables, and $p_j$ is the predicted probability of developing CAD for the $j_{\text{th}}$ individual.

The predicted incidence of CAD at the assessment level was calculated as the average of the predicted rates for all participants from the same assessment center. As well as in models with PRS and PCE only, GLMs with BMI and TDI are also tested. Section *Results for UKB assessment centers* shows that the standard GLM has a very poor fit at the assessment center level even though the individual-level prediction is acceptable [based on the Area under the curve (AUC) results]. The group-level prediction became even worse after a new variable was added into the model. A better model is, therefore, needed to predict the group incidence rates and we introduce one in section *The Mundlak model to predict the number at risk*.

### Exploration of performance using permuted groups

To identify potential reasons for the poor group-level predictions, we examined the performance of the same model on versions of the UKB data that had undergone a form of group label permutation. Specifically, we reallocated individuals to groups (assessment centers) at random, while preserving the group sizes (number of individuals in each assessment center) and approximate group-specific rates. We achieved this, along with an enforcement of correlation between PRS and nongenetic risk—mediated via the PCE risk—for both negative and positive correlation. We then examined the performance of the above GLM

(regressed only on PRS) as well as our proposed model below (also regressed on PRS terms only) and demonstrated that our proposed model outperforms the GLM, especially in the presence of negative correlation. The results for the permuted groups are reported in the section *Structured permutations results*. This result demonstrates the ability of GLM to distinguish between low and high incidence groups for the data set, in the presence of positive correlation between observed and unobserved covariates. We, therefore, speculated that some similar cryptic group structure might play a role in the poor fitting at the assessment center level, so we employed Pearson's correlation tests to reveal group structure. The cryptic group structure in the UKB data set is the reversed direction of the relationship of variables at the group level and the subgroup level. Such a scenario is well studied in the statistical literature and is referred to as Simpson's paradox. The section *Ascertainment bias confounds group rate estimation* demonstrates the presence of this phenomenon in our data set.

## The Mundlak model to predict the number at risk

We are interested in developing a model to predict CAD risk at the assessment center level without using the assessment center label, so that this model can be used to estimate the number of CAD risks for a new group where we only observe covariates, but not the group rate. Simple GLMs fit poorly at the assessment center level, and this poor fit is due to the opposite directional effect of the variables at the group level and the subgroup levels (see results in Ascertainment bias confounds group-rate estimation). Note that a latent variable model cannot be used to predict the group rate because it requires an estimate of the group rate, such as the observed rate in a sample, from which the group-specific intercept term can be estimated.

The Mundlak GLM fits our needs well. This model was originally conceived by Mundlak (1978) to analyze data consisting of repeated observations on economic units. In his model, group means of independent variables are included as predictor variables/covariates in addition to the original observed variables, so the assumption that observed variables should not be uncorrelated with unobserved variables is relaxed. Dieleman and Templin (2014) compared the random- and fixed-effects estimators (RE and FE, respectively) with the Mundlak GLM (called the within-between approach in this paper) for clustered data when unaccounted-for group-level characteristics affect the outcome variable. Even though RE and FE are commonly used competing methods in health studies, the Mundlak GLM outperforms those 2 estimators in their simulation study.

In this study, according to the GLM illustrated in Generalized linear model to predict the number of risk for regression on PRS and PCE, the Mundlak GLM simply adds group-mean variables into that model. We used the same approach as (Dieleman and Templin 2014):

$$\begin{aligned} \text{logit}(p_{gn}) = {} & \beta_0 + \beta_{\text{PRS}}(\text{PRS}_{gn} - \overline{\text{PRS}_g}) \\ & + \beta_{\text{PCE}}(\text{PCE}_{gn} - \overline{\text{PCE}_g}) \\ & + \gamma_{\text{PRS}}\overline{\text{PRS}_g} + \gamma_{\text{PCE}}\overline{\text{PCE}_g}, \end{aligned}$$

where $g \in (1 \ldots G)$ and $n \in (1 \ldots N)$ denote the group and observation identification within each group, respectively, and $G * N = J$ from Equation 1. For the $n_{\text{th}}$ individual belonging to the $g_{\text{th}}$ assessment center, $\overline{\text{PRS}_g}$ and $\overline{\text{PCE}_g}$ are the means of PRS and PCE for the $g_{\text{th}}$ assessment center. $\beta_0$ is the intercept, $\beta_{\text{PRS}}$ and $\beta_{\text{PCE}}$ are regression coefficients for the group demeaned PRS and PCE, respectively, and $\gamma_{\text{PRS}}$ and $\gamma_{\text{PCE}}$ are estimators for the corresponding group-mean PRS and

PCE. Here, Dieleman and Templin (2014) used the original variable minus the group mean as the input variables, rather than the original variables, for reasons explained in Bell and Jones (2015). According to Dieleman and Templin (2014), every $\beta$ represents the within-group effect and assesses changes within a group and every $\gamma$ measures the effect of the corresponding variable between groups.

## How the Mundlak model works

Results in section *Mundlak GLM results* show that the Mundlak model works well on prediction of CAD risk at the assessment center level. The reason for this significantly improved performance is that group-mean variables in the Mundlak GLM act as a proxy for unseen group-specific behavior, so the group structure can be captured in the Mundlak GLM. We next demonstrate a simulation experiment to better understand why the Mundlak GLM works.

The theory to support the simulation is that the risk of CAD increases with the increasing of PRS and PCE scores. We start the simulation with a reproduction of the Simpson's paradox scenario, using the same complete data set as detailed in section *Study flow*. Then we manually create groups based on which quantile 1 hidden variable falls in. We assume that this hidden variable $y_j \sim N(\mu_j, 1)$ is a random variable with mean value calculated as a linear combination of $PRS_j$ and $PCE_j$, so that:

$$\mu_j = \mathbb{E}[y_j] = \alpha_1 * PRS_j + \alpha_2 * PCE_j, \qquad (2)$$

where $\alpha_1$ and $\alpha_2$ are weights for variables PRS and PCE.

The weights, $\alpha_1$ and $\alpha_2$, are used to determine the existence and extent of Simpson's paradox. The severity of the reversed direction of the relationship of variables at the individual level and the group level can be controlled by the size of $\alpha_1$ and $\alpha_2$. For example, when $\alpha_1$ and $\alpha_2$ have the same signs, Y increases with increasing PRS and/or PCE. If we create several groups of equal size based on the ranked values of $y_i$ from lowest to highest, so that the first group contains samples with the lowest values of Y and the last group contains samples with the highest values, then the first group should have the lowest average PRS *and* lowest average PCE and the last group the highest average PRS *and* highest average PCE. When GLM is regressed on PRS and/or PCE, individuals with higher values of PRS and PCE should have higher probability of developing CAD. Similarly, comparing groups with increasing values of PRS and PCE, the group with high PCE and PRS values has more risky individuals than the group with low values. In this case, the individual level and the group level have the same CAD rate trend, so Simpson's paradox does not exist. However, predicted group rates will be biased toward the mean.

When the weights $\alpha_1$ and $\alpha_2$ have opposite signs, the relationship among groups is not as straightforward. For example, if we set $\alpha_1 = -0.5$ and $\alpha_2 = 0.5$, Y decreases with increasing PRS, but increases with increasing PCE scores. We also create equal-sized groups based on the ranked values of $y_i$ from lowest to highest. Under this scenario, the first group has the highest mean of PRS *and* the lowest mean of PCE, but the last group has the lowest mean of PRS *and* the highest mean of PCE. Because PRS and PCE contribute in opposite directions to group assignment, the CAD rates between groups will be less different than in the above scenario.

For the second scenario, because the risk of developing CAD depends on both PRS and PCE score in the same direction, GLM regressed on PRS and PCE will experience Simpson's paradox, which will lead to poor predictive performance at the group level. But the Mundlak GLM accounts for the opposite direction by finding individual level and group-level coefficients of opposite signs. Therefore, we expect a good fit of the GLM with the inclusion of group-mean variables. Section *Structured permutations results* confirms this expectation. We chose the values of $\alpha_1$, $\alpha_2$ in order to create groups with a similar spread of rates as the UKB assessment centers.

## Leave-one-center-out cross-validation

To assess out of sample performance, we applied the Mundlak model to the complete data set excluding 1 center at a time, and then applied the model to the data set from this excluded center. We called this method leave-one-center-out cross-validation (LOCOCV). We averaged the predicted values for this 1 center and obtained the predicted case rate for this center. After applying LOCOCV to all centers, we thus obtained a list of predicted case rates for each center, based on fitting the model to all other centers and the covariates. We then compared the performance of a standard GLM to an equivalent Mundlak model for different choices of predictor covariates. We used this LOCOCV method to assess model performance as it most closely resembles our motivating scenario wherein we trained a model on a set of groups and then used the trained model to predict or estimate rates in a new group for which we have covariate information but no observed case rate data. Furthermore, the Mundlak model could be prone to overfitting when the number of predictor covariates approaches the number of groups when training data includes the case statuses of the group being predicted,[1] whereas the LOCOCV procedure will address any upward bias in performance due to such overfitting.

## Confidence intervals for group rates

We used Wilson's score interval (Wilson 1927) to produce a 95% prediction interval for all models to quantify the uncertainty in the estimated incidence of CAD at the assessment center level. Wilson's score interval is used to construct confidence intervals for Binomial probabilities based on observations of $n_s$ successes out of $n$ trials. The maximum likelihood estimate is simply the mean $\hat{p} = n_s/n$ and Wilson's score interval improves the Wald interval, which is based on a Normal approximation to the sampling distribution for $\hat{p}$ to allow for asymmetry and the constraint that $0 \leq \hat{p} \leq 1$. Specifically, we used the fixed point iterative estimate given by:

$$p_g^{(k+1)} = p_g^{(k)} \pm z\sqrt{\frac{p_g^{(k)}(1-p_g^{(k)})}{n_g}},$$

for $k = 1, \dots$ until convergence and where $p_g^{(0)} = \sum_{j=1}^{n_g} \frac{p_j}{n}$ within each group $g$. Although this method was devised for constructing intervals for probabilities estimated from Binomial counts data, the same principles also apply to our usage here on group-mean probabilities estimated from multiple observations of estimated individuals' probabilities. Figure 8 shows that Wrexham has the widest confidence interval, followed by Swansea; these are the 2 mobile assessment units and have far lower sample sizes associated with them (see Table A1). We found that centers with predicted case rates close to observed case rates have relatively narrow prediction intervals.

---

[1] Essentially, the model would be sufficiently flexible to mimic having a latent group-specific term, thus predicting group-rates exactly.

# Results

## CAD events and rates

We extracted and compared the age of onset of the first CAD events between self-reported health outcomes with hospital inpatient data to determine the prevalence and incidence of CAD events. Supplementary Table S3 gives the number of first-ever CAD prevalence and incidence events from inpatient and self-reported records separately. Many participants reported CAD events in their self-reporting, but those events occurred too early to be recorded by inpatient data. This was consistent with suggestions from Eastwood *et al.* (2016) and Yeung *et al.* (2022), which both noted that using only UKB hospital inpatient data to identify prevalent cases would miss out many cases, as most prevalent cases were self-reported during the baseline assessment visit. Additionally, we found that the majority of participants with self-reported CAD events would have new CAD events recorded in their hospital inpatient records, with the majority occurring after they joined the UKB. Therefore, using only inpatient data would mistake actual prevalent cases as incidence cases. A total of 12 participants had only CAD events in their self-reported data and none in their hospital inpatient records, but no date of onset of CAD was given. We considered these participants as prevalent CAD cases. Thus, in conclusion, this study focus on CAD incident events, the complete data set used for analysis included a total of 263,087 UKB participants from 22 UKB assessment centers, with 8,458 participants developing CAD after enrollment in the UKB and the remaining 254,629 participants remaining free of CAD.

For all 22 assessment centers, we calculated age-standardized CAD prevalence rates on 2010 October 1 (the last day of attending assessment center for all UKB participants) and nonstandardized CAD incidence rates from 2010 October 1 to 2021 September 30 (the latest hospital inpatient record for CAD from our UKB file). Among all 22 centers, Wrexham and Swansea were mobile assessment centers. In addition, Stockport was a pilot center, which we have removed from our analysis. Figure 3(a) and (b) are maps for 21 UKB assessment centers with CAD rates created following steps explained in Appendix UKB location co-ordinates. To calculate the age-standardized CAD prevalence in 2010, 2 additional steps were taken in addition to following the definition of CAD definition to distinguish between prevalence and incidence of CAD events. Firstly, we removed UKB participants who died after enrollment in the UKB but before 2010 October 1, and secondly, we redefined incidence CAD events that occurred after participants enrolled in the UKB but before 2010 October 1 as prevalence events.

Figure 3(a) shows differences in the prevalence of CAD among centers. Cardiff and Bristol have the lowest CAD prevalence rates, whilst Wrexham and Glasgow have the highest rates. Figure 3(b) shows CAD incidence (without age standardization) for each assessment center. Stockport has the highest incidence rate, followed by Bury and Manchester. The numbers used to generate Fig. 3 are in Table A1.

## Complete data set

After distinguishing prevalence and incidence CAD events, calculating PCE scores for eligible UKB participants, extracting the CAD-PRS, BMI, and TDI provided by UKB, and filtering for samples with missing data, the complete data set had 263,087 participants, all of whom were White British. The detailed study flow chart is found in section *Study flow*. The overall data set had 3.21% incidence CAD event rate, with twice as many male patients as female patients. Summary statistics for risk factors used for PCE score

calculation for men and women in the complete data set are found in Table 1, which lists the summary statistics (mean, minimum, and maximum) for numerical risk factors and percentages for binary risk factors. In general, female participants had higher cholesterol levels, but lower levels of systolic blood pressure and BMI, and lower rates of smoking, hypertension medication, and diabetes. Table A1 provides the summary information for each UKB assessment center, including the number of UKB participants, the CAD incidence rates, and summary statistics for the risk factors assessed. The CAD incidence rates for each center in the complete data set are different from the corresponding incidence rates shown in Fig. 3(b), because UKB participants who were free of CAD or who reported as having a incidence CAD event but did not have complete information for PCE score calculation or CAD-PRS, were not included in the data set analyzed.

Figure 4 shows the density plots for PRS and PCE risk from the complete data set by CAD status and sex. Those plots show the ability of PRS and PCE risk to distinguish CAD cases and controls. The PRS density plots do not appear to differ between males and females, but samples with CAD from the complete data set have higher mean PRS values than samples without CAD.

## Statistical tests results

The permutation test was used to compare the PRS distribution between groups because it makes no assumptions about the distributions and can capture differences in the tails of the PRS distributions. We first applied this test on PRS values between samples from England and Scotland, but did not find any significant differences. We then applied this test across UKB assessment centers, and plotted *P*-value results in a heat map. Figure 5 classifies the *P*-values of permutation tests between any 2 assessment centers into 3 groups. It is not a symmetrical heat map due to sampling error, as a permutation test is a Monte Carlo resampling test. The smaller the *P*-value in Fig. 5, the higher the probability of a significant difference in PRS distribution between the 2 centers. For example, the distribution of PRS in Barts and Hounslow is different from many other assessment centers, but the distribution of PRS in Wrexham is not different from other locations.

From the complete data set, we also compared the distribution of PCE risk between the England and Scotland samples using the

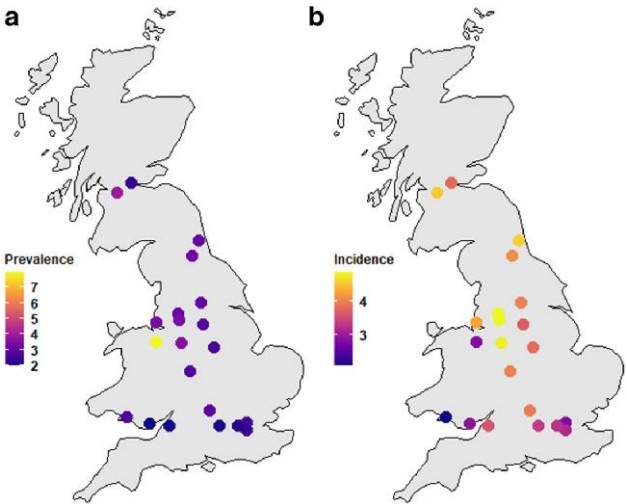

**Fig. 3.** Observed CAD rates of 22 UKB assessment centers:
a) Age-standardized CAD prevalence rates (as percentages), 2010 and
b) CAD incidence rates (as percentages), 2010–2021.

**Table 1.** Summary statistics of risk factors for the complete data set.

| Risk factors | Overall (N = 263, 087) | Males (N = 115, 150) | Females (N = 147, 937) |
|---|---|---|---|
| Incidence CAD events | 8,458 (3.21%) | 5,870 (2.22%) | 2,588 (0.99%) |
| Age joined UKB | 55.9 (39, 73) | 55.8 (39, 73) | 55.9 (40,70) |
| Total cholesterol (mg/dl) | 226.7 (130, 320) | 222.5 (130,320) | 230.0 (130,320) |
| HDL cholesterol (mg/dl) | 56.8 (20.3,100) | 50.4 (20.3,99.9) | 61.8 (20.3,100) |
| Systolic blood pressure (mm Hg) | 137.2 (90,200) | 140.7 (90,200) | 134.5 (90,200) |
| Smoking status (%) | 43.0% | 47.6% | 39.4% |
| Hypertension medication % | 13.4% | 13.9% | 13.1% |
| Diabetes mellitus (%) | 1.3% | 1.8% | 1.0% |
| Body mass index | 27.0 (12.1, 66.2) | 27.4 (12.8, 61,7) | 26.7 (12.12, 74,7) |
| Townsend deprivation index | Calculated immediately prior to participant joining UKB. Based on the preceding national census output areas. | | |

Risk factors in italics are parameters included in the calculation of PCE risk. HDL, high-density lipoprotein cholesterol.

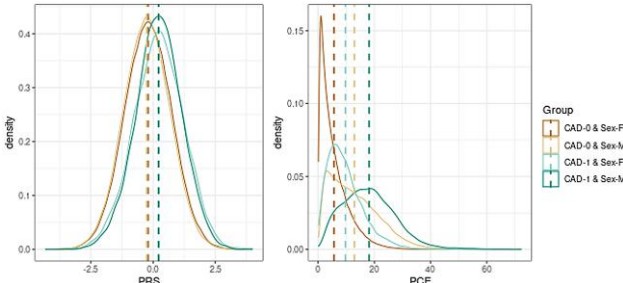

**Fig. 4.** Density plots for PRS and PCE risk from the complete data set. CAD-0, samples without incidence CAD events; CAD-1, samples with incidence CAD events; F, female; M, male. The dashed lines are the mean values for each variable.

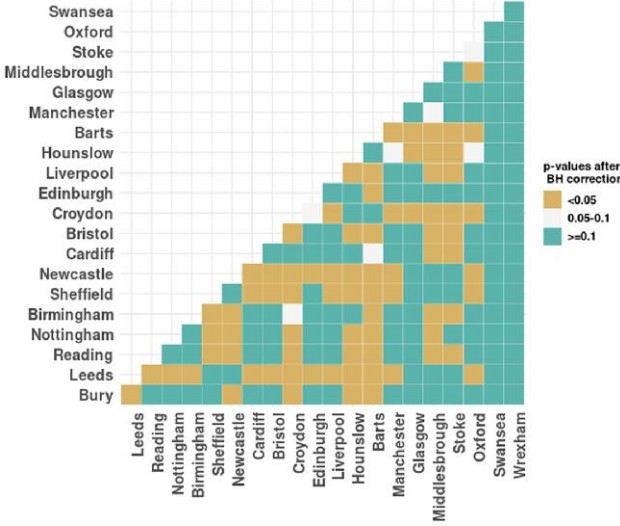

**Fig. 5.** Permutation tests on PRS across all pairs of UKB assessment centers. The BH correction is applied to control the false discovery rate. "<0.05" denotes adjusted *P*-value < 0.05; "0.05–0.1" denotes adjusted *P*-value between 0.05 and 0.1; "≥ 0.1" denotes adjusted *P*-value ≥ 0.1.

Mann–Whitney test (Yang *et al.* 2021). We found significant distribution differences of PCE risk (*P*-value = 3.985e − 05), rather than the small statistically significant differences found by (Yang *et al.* 2021) (*P*-value = 0.009) on the distribution of FRS and QRISK3 between England and Scotland. We then used the permutation tests on PCE risk across the UKB assessment centers, and the results are shown in Fig. 6. Figure 6 shows that, with the exception of Wrexham, the PCE risk distributions of all the other centers are very different from each other.

## Results from standard GLMs

### Results for UKB assessment centers

We used GLMs regressed on selected variables to predict the probability of developing CAD for each sample in the complete data set, and then calculated the assessment-level incidence of CAD as the mean of the predicted rates for all participants in the same assessment center. Figure 8 plots the relationship between the observed case rates and the predicted cases rate regression on PRS only and Table 2 gives the corresponding AUC from that GLM and the correlation between observed and predicted group rates for multiple models. In general, the AUC is relatively high, especially when the PCE score is used independently. When we started with PRS and gradually added more variables in the model, the AUC increased slowly and all variables exhibited a significant positive relationship with the risk of CAD.

However, the prediction of the case rate at assessment center level is very poor, as the predicted case rates for all assessment centers are very close in all GLMs. The relatively high correlation of the PRS GLM is due to the fact that the predicted rate increases with the observed case rate, but the PRS line in Fig. 8 shows that the predicted case rates remain very close across centers. We

also tested GLMs with interaction and quadratic terms, but did not obtain better performance than for GLMs with only linear variables. The poor performance of the standard GLM in predicting group rates as the mean of the individuals' data could be due to lack of model fit at the individual level, or due to some form of group structure not accounted for in the model. To identify which of these categories the issue arises from we next create random groups with varying rates from the real UKB data and demonstrate a much improved fit at the group level. Thus the issue is one of cryptic group structure, which we account for using a Mundlak model. An intuition behind why such an approach works is apparent from the counter-intuitive result seen in Table 2 wherein adding Age as a covariate actually deceases the LOCOCV correlation to a negative value. This is because although both PRS and age are both positively correlated with CAD, they are negatively correlated with each other (see Fig. 7). We investigate this phenomenon further in the next section.

### Ascertainment bias confounds group-rate estimation

We compared the correlation between 2 risk factors of the complete data set and at the group level using Pearson's correlation. First, we calculated the correlation between each pair of variables for the complete data set following the flow chart detailed in

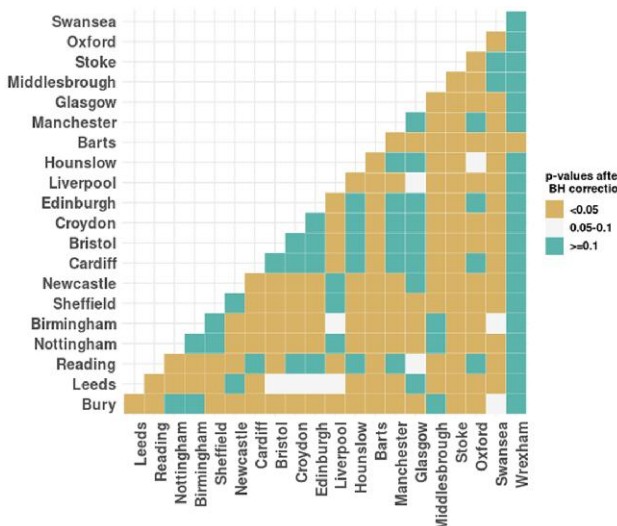

**Fig. 6.** Permutation tests on PCE risk across UKB assessment centers. The BH correction is applied to control the false discovery rate. "<0.05" denotes adjusted *P*-value < 0.05; "0.05–0.1" denotes adjusted *P*-value between 0.05 and 0.1; "≥ 0.1" denotes adjusted *P*-value ≥ 0.1.

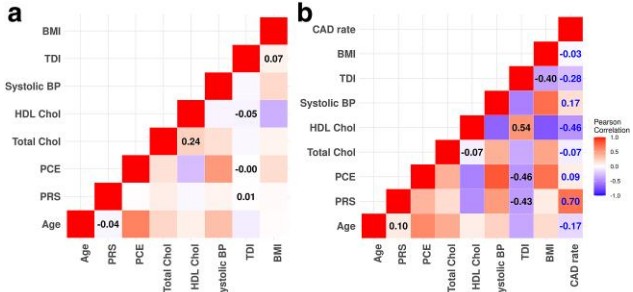

**Fig. 7.** The Pearson correlation a) across all individuals independent of assessment center and b) using mean values from each of the UKB assessment centers, including the mean observed CAD rate. Correlation coefficients are added in the boxes for pairs of variables where the signs of the coefficients are opposite between a) and b). PCE, pooled cohort equation, PRS, polygenic risk score; HDL, high-density lipoprotein cholesterol; BP, blood pressure; Total Chol, total cholesterol; BMI, body mass index; TDI, Townsend deprivation index.

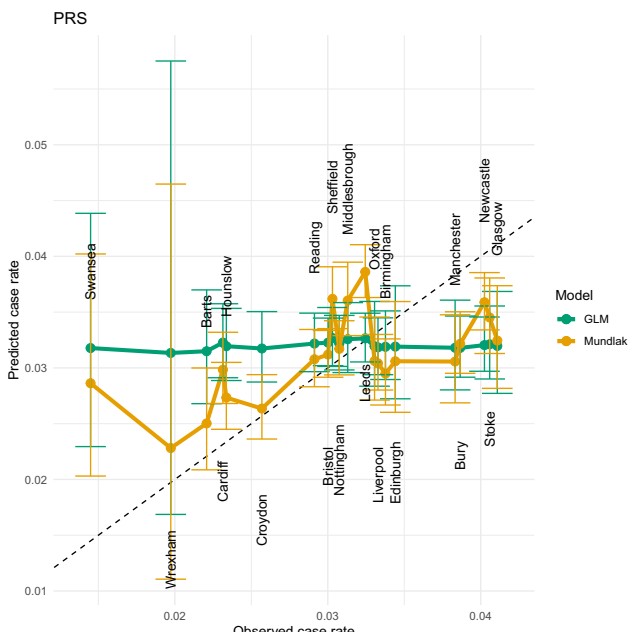

**Fig. 8.** LOCOCV predicted case rates from a GLM (green) and a Mundlak GLM (orange), where both are regressed on PRS only. Observed case rates are CAD incidence rates for each UKB assessment center and predicated case rates are the mean of the predicted rates for all participants in the same center.

section *Study flow* and plotted the coefficients in a heat map [Fig. 7(a)]. For example, the coefficient between PRS and PCE risk was calculated using values from all 263,087 samples in the complete data set as 0.009. We then calculated the correlation between each pair of variables using the mean of the assessment center for each variable. There are 21 assessment centers in the complete data set—we calculated the mean of each assessment center for each variable. Using the same example, we had 21 PRS means and 21 PCE risk means, we then calculated Pearson's correlation using these 2 sets of means. In this case, the correlation between PRS and PCE risk was 0.39, much higher than the previous value. Only PCE and TDI were not statistically significantly correlated at the individual level after adjustment for multiple comparisons using BH adjustment at the 0.01 level. At the level of the assessment center, there were 10 pairs of covariates that were statistically significantly correlated at the 0.05 level, after controlling for multiple comparisons. This is not surprising, given that there are only 21 assessment centers so $n = 21$ for all such comparisons and there were $\binom{9}{2} = 36$ comparisons being made in Fig. 7. The significantly correlated covariate pairs are: (age + PCE), (PCE + HDL), (PCE + SBP), (HDL + SBP), (HDL + TDI), (SBP + TDI), (PCE + BMI), (HDL + BMI), (SBP + BMI), (PRS + CAD). Interestingly, only PRS is statistically significantly correlated with CAD rate at the assessment center level.

The inverse relationship between 2 variables at different levels is a well-known phenomenon, termed Simpson's paradox (Pearl 2014). Another example in this study is the relationship between the variables PRS and TDI. We can see that their correlation is negative at the assessment center level, but slightly positive for the complete data set. This reversed relationship can also be found for continuous variables included in the PCE risk calculation. For example, Fig. 7 shows that such a phenomenon exists between PRS and age, between PRS and TDI and between systolic blood pressure and TDI. One possible reason for the inverse relationship at the individual level and group level is participation bias (Schoeler *et al.* 2023). For example, if a group has a higher rate of death from CAD for some reason (e.g. higher average age), then the group mean of the surviving people from whom a sample can be taken will have a lower PRS based risk, despite

the homogeneity of genetics between the groups before people died off. This creates an ascertainment bias that varies between groups.

## Mundlak GLM results

After identifying the potential cause for the poor fit at the assessment center level, we employed the Mundlak GLM to deal with this problem. The Mundlak GLM in this study was built based on GLMs regressed on the original variables as well as the group means of the same variables. Figures 8–11 show a very strong positive relationship between the observed case rates and the predicted case rates for all 21 assessment centers after including the group-mean variables in the GLMs. The AUCs from Mundlak GLMs are given in Table 2, where the values are close to—but

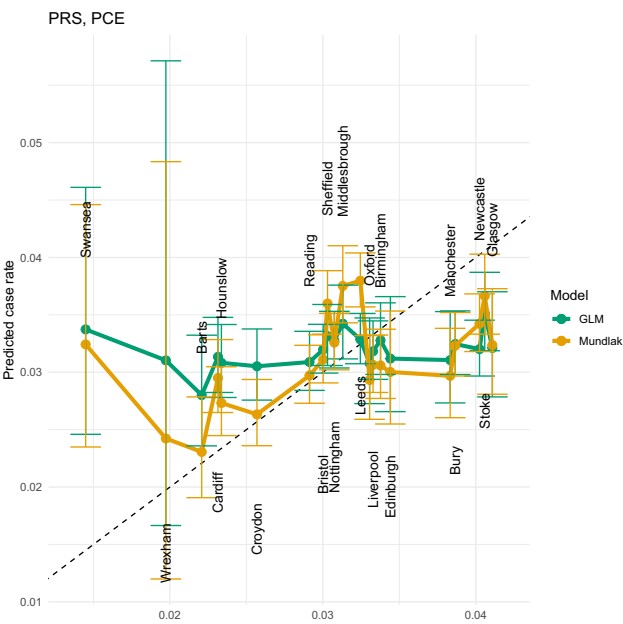

**Fig. 9.** LOCOCV predicted case rates from a GLM (green) and Mundlak GLM (orange), regressed on both PRS and PCE values.

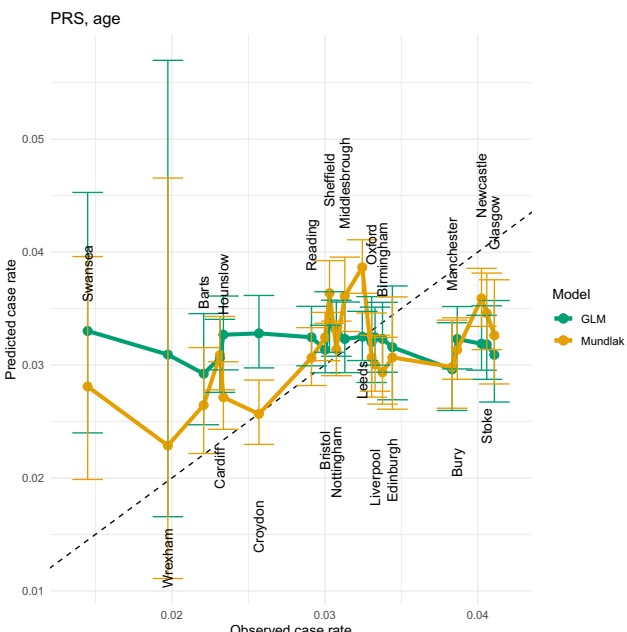

**Fig. 10.** LOCOCV predicted case rates from a GLM (green) and Mundlak GLM (orange), regressed on PRS and Age. We see a similar performance in LOCOCV to the PRS + PCE results in Fig. 9, indicating that age is the primary explanatory variable in PCE.

uniformly higher than—the AUCs for the corresponding GLMs. More importantly, the correlations between observed and predicted group rates from the Mundlak GLMs in Table 2 are higher than those for the GLM models. This means that compared with standard GLMs, the Mundlak GLMs significantly improved the prediction accuracy at the assessment center level. To calculate these values we used out-of-sample cross-validation; see section *Leave-one-center-out cross-validation for details*, but notably even for in-sample prediction wherein the model is fit using all centers'

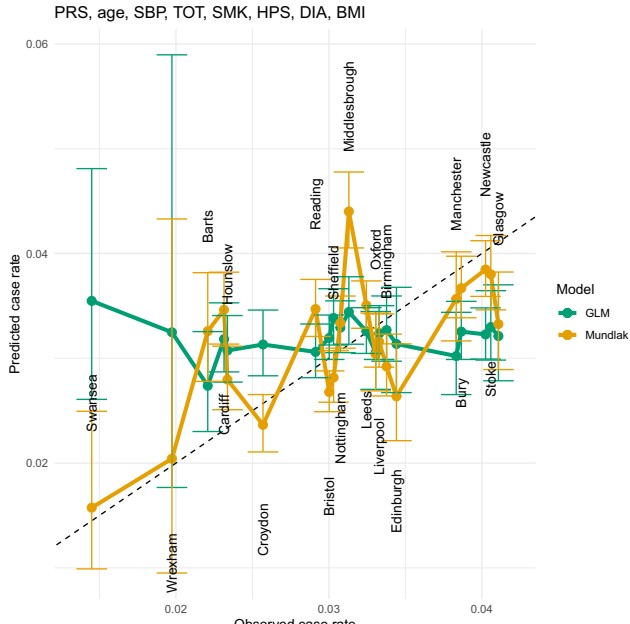

**Fig. 11.** LOCOCV predicted case rates from a GLM (green) and Mundlak GLM (orange), regressed on PRS, Age, SBP, TOT, SMK, HPS, DIA, and BMI. PRS, polygenic risk score; SBP, systolic blood pressure; TOT, total cholesterol; SMK, smoking status; HPS, hypertension status; DIA, diabetes status; BMI, body mass index.

data the standard GLM performs poorly in terms of group-rate prediction.

The Swansea assessment center is a notable outlier. A possible reason for this is that this center has a larger number of older participants, as this center has the highest average age of any center in the complete data set (see Table A1). Nanna *et al.* (2020) examined the performance of PCE in older adults, and found poor performance of PCE for ASCVD risk estimation in older adults. This center has the highest mean value of PCE risk, but the lowest CAD incidence rate (see Table A1) and this phenomenon reduces the accuracy of the assessment center level predictions. The relatively low mean PRS value in Swansea compared with other centers is consistent with its lower incidence of CAD, which also explains why the Mundlak GLM regressing on only PRS gives almost the closest predicted case rate to the observed case rate vs other models. Manchester and Glasgow are another 2 outliers and both centers have relatively higher CAD rates, but relatively low mean ages and PCE risk. One possible reason for the poor fit of Manchester and Glasgow can be explained by the high *P*-values from the permutation results shown in Fig. 5. The distribution of PRS in Manchester and Glasgow is not significantly different from PRS in other centers, which makes forecasting more difficult. Table 3 lists the estimated coefficients for this model along with standard errors.

Instead of using the PCE risk directly, we also tested the Mundlak GLM models on the PRS and the risk factors used in the PCE risk calculation. There are 10 variables in total, including 7 variables from PCE, PRS, BMI, and TDI. We started by running Mundlak GLM on only 1 of the 10 variables and selecting the model with the highest AUC, then we ran the Mundlak GLM on 2 variables [with $10_2 = 45$ nonrepeating combinations] and selected the 2-variables Mundlak GLM with the highest AUC. The same steps were repeated, adding only 1 new variable each time, until all 10 variables were included in the Mundlak GLM. Only linear combinations were included in the Mundlak GLM, because we

**Table 2.** AUC for GLMs and Mundlak GLMs regressed on different sets of covariates, trained and tested on the same complete data set, and the correlation between observed and predicted group rates for LOCOCV.

| Covariates included | GLM AUC | Mundlak AUC | GLM LOCOCV | Mundlak LOCOCV |
| --- | --- | --- | --- | --- |
| PRS | 0.632 (0.626, 0.638) | 0.635 (0.629, 0.641) | 0.251 (−0.203, 0.616) | 0.619 (0.256, 0.829) |
| PRS + PCE | 0.752 (0.746, 0.757) | 0.753 (0.748, 0.758) | 0.304 (−0.147, 0.65) | 0.498 (0.084, 0.765) |
| PRS + age | 0.69 (0.685, 0.696) | 0.693 (0.687, 0.698) | −0.053 (−0.474, 0.388) | 0.585 (0.206, 0.812) |
| PRS + age + HDL + SBP | 0.74 (0.735, 0.745) | 0.741 (0.736, 0.746) | 0.289 (−0.163, 0.64) | 0.531 (0.129, 0.783) |
| PRS + age + SBP + TOT + SMK + HPS + DIA | 0.72 (0.714, 0.725) | 0.722 (0.717, 0.727) | 0.0452 (−0.394, 0.468) | 0.619 (0.255, 0.829) |
| PRS + age + SBP + TOT + SMK + HPS + DIA + BMI | 0.722 (0.717, 0.727) | 0.724 (0.719, 0.73) | −0.0101 (−0.44, 0.423) | 0.655 (0.311, 0.847) |

In brackets are 95% CIs, computed for the AUC using the DeLong *et al.* (1988) method to estimate variance, followed by a Normal quantile calculation. PCE, pooled cohort equation; PRS, polygenic risk score; HDL, high-density lipoprotein cholesterol; SBP, systolic blood pressure; TOT, total cholesterol; SMK, smoking status; HPS, hypertension status; DIA, diabetes status; BMI, body mass index.

**Table 3.** Estimated Mundlak GLM coefficients along with standard errors, using demeaned individual observations and group-mean variables.

| | |
| --- | --- |
| (Intercept) | −3.69***(0.44) |
| PRS | 0.52***(0.01) |
| PCE | 0.08***(0.00) |
| Group mean PRS | 3.90***(0.57) |
| Group mean PCE risk | 0.09 (0.04) |
| Num. obs. | 2,63,087 |

\*\*\*$P < 0.001$.

checked that models with interactions or quadratic terms did not improve the predictive performance. We tested a total of 1,023 Mundlak GLMs, and presented the results of selected models in Figs. 8–11 and Table 2. Among all 10 variables, PRS has the best prediction power, followed by age and HDL cholesterol. AUC values for Mundlak GLMs in Table 2 are uniformly higher for GLMs. To address any concerns around potential over-fitting due to the high number of regressors, we assessed our models using leave-one-group-out cross-validation in section *Leave-one-center-out cross-validation*.

We see that including PCE with PRS achieves only a modest improvement relative to regression only for the GLM in terms of LOCOCV, a notable improvement in AUC for both the GLM and Mundlak GLM, and a disimprovement in terms of LOCOCV correlation for the Mundlak GLM. Separately including the covariates which PCE comprises achieves a good balance between individual-level performance in terms of AUC and group-level prediction in terms of LOCOCV correlation.

## Structured permutations results

To better understand why the Mundlak model gave much better predictive performance at the assessment center level, we designed a simulation experiment as described in section *How the Mundlak model works*. Following the permutation design, we assumed that there was a hidden random variable with mean value calculated as a linear combination of PRS and PCE risk and then manually created 21 groups based on which quantile the hidden variable fell into.

If $\alpha_1$ and $\alpha_2$ from Equation 2 were assumed to have the same sign, the group with fewer CAD events should have lower values of both PRS and PCE risk. We set $\alpha_1 = \alpha_2 = 0.2$ and then compared the performance of the standard GLMs when regressed on PRS only and the performance of the Mundlak GLM when regressed on PRS and group-mean PRS. The orange and green lines from Fig. 12(a) show that the Mundlak model has much better group-level prediction than the GLM. This is because, in the simulation setting, the observed case rate was determined by both PRS and PCE risk, so regressing on PRS alone could not predict the case

rate well. When only PRS was included in the Mundlak model, the dependence of CAD on PCE could be captured by the group mean of PRS, so the Mundlak model performs much better than a standard GLM. The group mean of the variable acts as a proxy for unseen group-specific behavior in the Mundlak model.

If $\alpha_1$ and $\alpha_2$ from Equation 2 are assumed to have opposite signs and the groups were still determined by the hidden variable in Equation 2, there is no simple relationship between the severity of CAD risk and the value of PRS and PCE risk. We set $\alpha_1 = −0.5$ and $\alpha_2 = 0.5$ and let the GLM and Mundlak GLM both be regressed on PRS only. In this setting, the groups were determined by the opposite direction between PRS and PCE risk, but the risk of CAD was dependent on both variables in the same direction, so regressing CAD only on PRS experiences the Simpson's paradox. Figure 12(b) shows the results from the GLM (green line), the Mundlak GLM (orange line). The GLM actually predicts low case rates for groups with high observed case rates and predicts high case rates for groups with low observed case rates, whereas the Mundlak GLM can reveal the hidden inverse relationship between PRS and PCE risk, because group-mean PRS acts as a proxy for the unseen relationship. Appendix Influence of weights on within-group correlation in structured permutations discusses details of the influence of $\alpha$ values on within-group correlation between PRS and PCE.

## Discussion

We proposed a framework for estimating the CAD case rate or number at risk in a homogeneous group of people, based on combining genetic and nongenetic contributions to risk. We demonstrated that simply fitting a logistic regression to the UKB and then estimating group rates as the average predicted probability of CAD in the target sample has exceptionally poor performance. We showed that this is largely attributable to a reversal of correlation between genetic and nongenetic risk factors at the group or cohort level compared to the correlation at individual level. Such behavior manifests as an example of Simpson's Paradox wherein, for example, PRS and TDI are positively correlated across participants at the individual level, but the group-specific mean values are negatively correlated. This can occur due to ascertainment bias, also known as participation bias or collider bias.

Population-based cohort studies, including the UKB study, are subject to participation bias. Fry *et al.* (2017) compared the socio-demographic and health-related characteristics of UKB participants with the general population and found that the UKB participants were more likely to be older, female, and wealthier. Weng *et al.* (2019) compared the TDI gathered from 8,848 households in the 2001 UK Census and the 502,625 participants in the UKB cohort and found that UKB participants were generally less

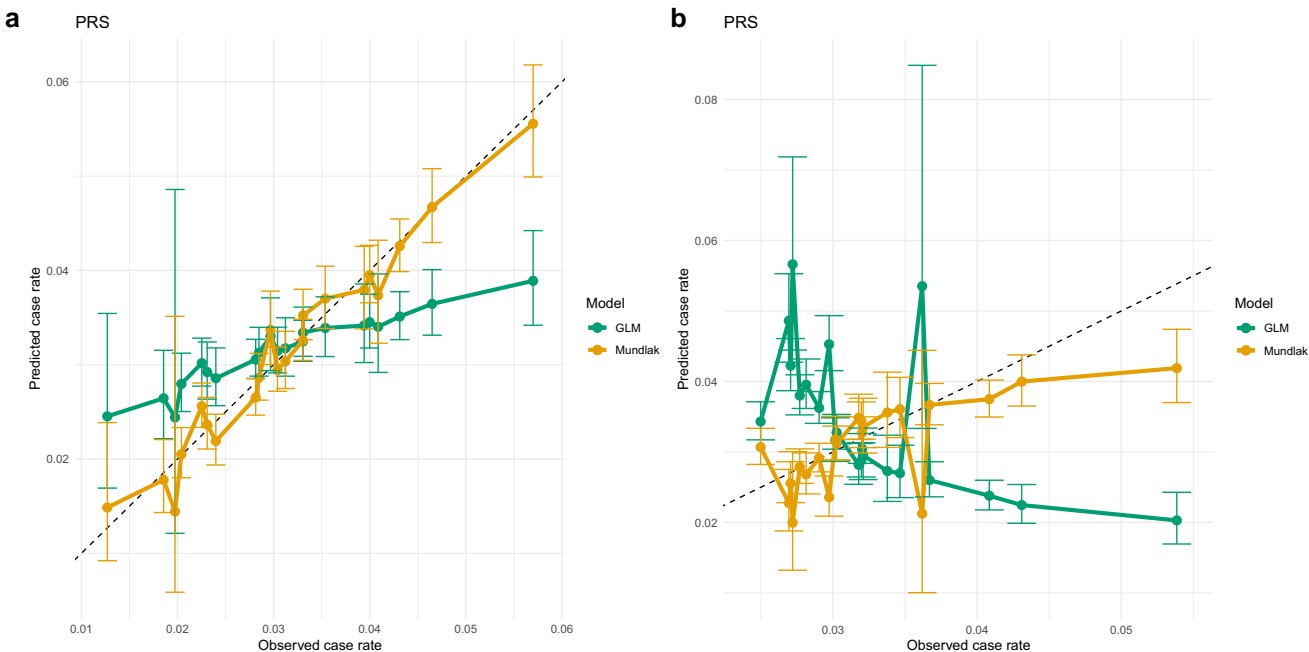

**Fig. 12.** LOCOCV predicted case rates for permuted groups, a) when $\alpha_1$ and $\alpha_2$ from Equation 2 have the same sign (both positive) and b) when $\alpha_1$ and $\alpha_2$ from Equation 2 have opposite signs.

deprived than the general UK population. Schoeler *et al.* (2023) demonstrated that the selective participation of the UKB cohort twisted the genome-wide associations and genetic correlation results compared with results in probability samples. Our study showed that participation bias altered risk prediction at the group level. Using the same UKB data, the effect of population structure in different geographical areas has been studied by Lin *et al.* (2022) on the estimation of SNP heritability. Our study discussed the effect of PRS in different regions.

An example of a cause of such bias is where 2 groups of initially similar PRS distributions experience different CAD rates due to an unobserved or lurking variable such as a differing age profiles or lifestyle factors. Since PRS also contributes to risk, the higher death rate in 1 group will lead to the survivors having lower average PRSs as more of those with higher polygenic risk will have died and been removed as candidates for a sample in that group. Thus the samples from the 2 groups will have lower polygenic risk in the group with higher case rates. The direction of the relationship between polygenic risk and disease status is then reversed at the group (as opposed to individual) level.

We showed that this source of bias exists within the UKB when using the assessment centers as groups, but then showed how to account for this structure using a Mundlak model wherein group specific means of covariates are included in the regression model. We demonstrated that such an approach has the ability to predict individual-level disease status, with an accuracy that is improved relative to a model without such terms. But more importantly, it has much improved ability to estimate the number at risk or case rate of a prospective group using samples for which disease status is unknown, but the regression covariates are available. UKB assessment centers have been used to adjust for bias in statistical analysis. For example, Lu *et al.* (2022) conducted GWAS and constructed PRS using UKB data, adjusting for terms such as age, sex, and recruitment center in their models.

Recent research (Lin *et al.* 2023) (using UKB data) shows that factors such as age, sex, genetic batch, and assessment center

potentially exert a greater influence on PRS predictions compared to the inclusion of PCs in the presence of genetic heterogeneity. Our study shows that adding group-specific means of covariates can also improve prediction at the individual level, while greatly improving the model's ability to estimate group-specific rates based on samples of genetic data and other covariates.

In a recent study on genetic correlates of social stratification, with a focus on SES, Abdellaoui *et al.* (2019) found that "many of the regional differences in health-related outcomes are not likely to be caused by differences in genetic variants that are directly causal for the health outcomes themselves, and are more likely to be due to environmental factors correlated with regional SES and therefore also with the SES-related genes of inhabitants." In other words, for these SES phenotypes, our Mundlak model may work well precisely because it models out this correlation between gene and environment. In this paper, we have demonstrated this ability for CAD. We noted the similarity between our model and one of those assessed in Abdellaoui *et al.* (2022). In that paper, it was shown that the inclusion of a regional-average PRS term improved SES phenotype prediction in a siblings-based study using UKB data. They assessed this model only at the level of the individual and show that incorporating such a regional-level correction leads to a boost in accuracy of their regression model. We also observe a modest but uniform increase in AUC under the Mundlak model vs the standard GLM for all combinations of covariates, but this is simply attributable to our model having additional predictors.

Most compelling of all is our result that the Mundlak regression model performs consistently well on out-of-sample group-rate predictions as evidenced by the leave-one-group-out cross-validation. This demonstrates that the ascertainment (or participation or collider) bias that causes the individual-level logistic regression model to perform poorly in group-rate predictions is reduced in a systematic, consistent, and appropriate manner across assessment centers. This, in contrast to a latent variable or mixed model with group-specific intercept terms, can be used

to predict group rates based on new samples without an existing and accurate estimate of disease case rates. In our simulation experiment, we have shown that the Mundlak model can reveal the hidden inverse relationship between PRS and PCE risk even when only PRS was included in the model. This suggests that the Mundlak model has the potential to make accurate predictions when there is a significant variable that determines the risk but cannot be incorporated into the model directly.

Commercial genetic testing services have been sold more than 27 million times, but the ability of genetic factors to assess risk did not outperform common methods for CAD (van Dam *et al.* 2023). They also pointed out that risk assessment for CAD based on simple questionnaires or variables from electronic health records is as good or better than risk prediction based on genetics alone. For this reason, they recommended continuing to use questionnaire techniques for initial risk assessment rather than relying on genetic testing alone to determine risk. Our results suggest that for commercial providers of genetic testing services, prediction at the individual level can be significantly improved by adding group-mean variables to the risk prediction model, and that age is a relatively easily obtained group indicator.

This study has limitations. The first limitation is that we use the group-specific means of the same variables that are used in standard GLMs to adjust for the ascertainment bias. The group means of the independent variables may not fully capture the ascertainment bias between centers, as there may be other characteristics at the assessment center level that affect the outcome, but that we have not included in our analysis. As health facilities in a single geographical area may share budgets, Dieleman and Templin (2014) noted that other sources may introduce ascertainment bias to health facilities, including guiding policies, attitudes toward treatment, population, disease patterns, and supply constraints. For the UKB assessment centers, the original function of each assessment center (for example, whether it is a clinic or a hospital), is another possible characteristic. If we had more center-specific variables to add to this model, it might help explain more of the variation. Fortunately, if any such unseen factors are in any way correlated with any variables we do include at the group level, then the Mundlak model will account for them, up to that level and correlation.

Additionally, we only test the Mundlak model on the UKB participants, not on other external data sets. Single ancestry basis is another limitation of this study, as the complete data set only includes White British. Many studies have called for an increase in diversity in large-scale genetic association studies (e.g. Duncan *et al.* 2019; Schoeler *et al.* 2023). Also, the accuracy of disease risk prediction was shown to improve after adding family history to the model (Gim *et al.* 2017), but this study did not explore the effect of family history on the group structure or the effect of other risk factors from the UKB resources. This can be investigated in future studies.

In conclusion, We distinguished prevalence and incidence CAD events for all UKB participants and identified geographical variations in CAD age-standardized rates across UKB assessment centers. The standard CAD-PRS provided by the UKB resources was selected to represent the genetic risk, as this set of PRS had the best predictive performance. We calculated PCE risk to represent the nongenetic risk factors for CAD. There were significant distributional differences in PRS and PCE risk between UKB participants from England and Scotland, according to the results of the Mann–Whitney test. Permutation test results showed that PRS from different assessment centers differed significantly. The group-level predictive performance of standard GLMs was biased by a reversal of the correlation between genetic and nongenetic risk factors at the group or cohort levels, compared to the individual level. This behavior was effectively modified by the Mundlak model, which included the group-specific means of covariates along with the original covariates in GLMs. The group means of the covariates acted as a proxy for the unobserved group-level characteristics that affected the outcome variables. The Mundlak model has the advantage of predicting the number at risk in a new group, given a sample of individual-level data. We showed that our model can effectively predict case rates in out-of-sample groups even in the presence of ascertainment bias that confounds group rate estimation. Our method corrects for systematic biases at the cohort level and has potential applications in public health planning, including screening programmes and early intervention strategies.

## Data availability

The study analyses were based on data from the UK Biobank website (http://www.ukbiobank.ac.uk). UK Biobank data are open source and available to researchers following acceptance of a research proposal and payment of an access fee. This research has been conducted using the UK Biobank Resource under Application Number 59528.

Supplemental material available at GENETICS online.

## Funding

This publication has emanated from research conducted with funding from the Science Foundation Ireland under Grant number (SFI/12/RC/2289_P2). For the purpose of Open Access, the author has applied a CC BY public copyright licence to any Author Accepted Manuscript version arising from this submission.

## Conflicts of interest

The author(s) declare no conflicts of interest.

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

*Editor: N. Zaitlen*

# Appendix A

## CAD rates and average covariate values for assessment centers

Table A1 shows the CAD prevalence and incidence rates for all assessment centers, including the pilot center at Stockport which we do not include in our analyses. There are individuals for which PCE or PRS values are not available, hence we also report the rate of CAD rates (as percentages) once we retain only those individuals within each center that we have complete information for. The number remaining in each assessment center is provided, along with mean age, mean PRS, mean PCE, mean BMI, and mean TDI values for each center.

## Results for age groups

Splitting groups by age is common in health-related studies, so we repeated our analysis using age as the group indicator. Figure 7 shows that PRS has a strong negative correlation with age. This is because participants with highly elevated PRS had developed CAD or other diseases, so they do not show up in the complete data set, confirming the existence of collider bias. As there were only few participants in the age group [35,39], we excluded this group from the analysis.

Figure A1 shows the predicted case rates plotted against the observed case rates from the standard GLMs and the Mundlak GLMs. When the PCE risk is included in the regression model, both the individual and group-level perform well as age is included in the calculation of the PCE risk. Table A2 shows that the standard GLMs and the Mundlak GLMs have similar levels of AUC. The advantage of the Mundlak model is evident when regressing only on PRS [Fig. A1(a)], as the predicted case rates are close to the observed case rates, but not in the standard GLM.

Comparing the GLM and the Mundlak GLM regressed on PRS, the Mundlak GLM has a better risk classification performance, with an net reclassification improvement (NRI) of 3.54% (95% CI, 2.12%–4.92%). This result is similar to the NRI obtained by Elliott *et al.* (2020) by comparing the model with PCE and PRS with the model with PRS only.

van Dam *et al.* (2023) showed that the incidence in the 10% most at risk group of individuals increased from 2.4-fold and 3-fold to 4.7-fold risk for CAD by including common risk factors in the model with PRS only. Our results showed that the incidence in the 10% most at risk group of individuals increased from 2.3 (95% CI, 2.1–2.5) to 2.9 (95% CI, 2.7–3.1) times the risk of CAD by including the group-mean PRS in the model with PRS only.

## UKB location co-ordinates

UKB provides the grid coordinates for all assessment centers (UKB Resource 11,002). These grid coordinates are not latitude and longitude information, but figures obtained from the Ordnance Survey National Grid geographical reference system, whose measurements are easting and northing with a reference point near the Isles of Sicily (UK Biobank: deriving the grid coordinates). We first translated the UKB grid coordinates of the UKB into latitude

**Table A1.** Observed CAD rates for 22 UKB assessment centers, in order of low to high.

| Center | Age-standardized 2010 Prevalence (%) | 2010–2021 Incidence (%) | Total number | CAD number | CAD rate (%) | Mean age | Mean PRS | Mean PCE | Mean BMI | Mean TDI |
|---|---|---|---|---|---|---|---|---|---|---|
| Swansea (mob) | 2.743 | 2.102 | 1,104 | 16 | 1.449 | 56.605 | −0.235 | 9.847 | 27.754 | −0.702 |
| Wrexham (mob) | 7.994 | 2.844 | 304 | 6 | 1.974 | 55.661 | −0.287 | 9.032 | 28.321 | −1.846 |
| Barts | 2.539 | 2.794 | 4,529 | 100 | 2.208 | 54.586 | −0.267 | 7.847 | 25.836 | 4.07 |
| Cardiff | 1.958 | 2.9 | 10,932 | 253 | 2.314 | 55.148 | −0.23 | 8.709 | 27.655 | −2.134 |
| Hounslow | 2.114 | 3.285 | 11,383 | 266 | 2.337 | 56.249 | −0.251 | 8.741 | 26.285 | −0.674 |
| Croydon | 2.378 | 3.193 | 11,871 | 305 | 2.569 | 56.497 | −0.254 | 8.79 | 26.599 | −0.816 |
| Reading | 2.105 | 3.358 | 17,405 | 507 | 2.913 | 56.007 | −0.22 | 8.627 | 26.595 | −3.054 |
| Bristol | 1.993 | 3.591 | 26,415 | 793 | 3.002 | 55.263 | −0.217 | 8.827 | 26.705 | −2.395 |
| Sheffield | 2.841 | 3.712 | 17,455 | 529 | 3.031 | 56.326 | −0.188 | 9.144 | 27.12 | −1.708 |
| Nottingham | 2.627 | 3.824 | 20,031 | 616 | 3.075 | 56.337 | −0.212 | 9.232 | 27.01 | −2.064 |
| Middlesbrough | 3.303 | 4.094 | 12,330 | 386 | 3.131 | 55.744 | −0.187 | 9.385 | 27.387 | −1.707 |
| Leeds | 3.159 | 3.998 | 25,454 | 826 | 3.245 | 55.802 | −0.179 | 8.976 | 27.066 | −1.903 |
| Oxford | 2.968 | 3.95 | 8,258 | 273 | 3.306 | 55.875 | −0.219 | 8.511 | 26.394 | −2.21 |
| Liverpool | 3.434 | 4.294 | 18,268 | 608 | 3.328 | 56.157 | −0.219 | 9.117 | 27.398 | −1.352 |
| Birmingham | 2.86 | 4.01 | 12,555 | 424 | 3.377 | 56.046 | −0.226 | 9.305 | 27.335 | −1.109 |
| Edinburgh | 2.323 | 3.804 | 4,649 | 160 | 3.442 | 55.7 | −0.219 | 8.703 | 26.567 | −1.937 |
| Manchester | 3.445 | 4.847 | 7,332 | 281 | 3.833 | 54.643 | −0.218 | 8.636 | 26.987 | −1.118 |
| Bury | 3.205 | 4.857 | 15,754 | 609 | 3.866 | 56.217 | −0.206 | 9.329 | 27.309 | −1.92 |
| Newcastle | 3.097 | 4.603 | 20,152 | 811 | 4.024 | 55.791 | −0.179 | 9.001 | 27.333 | −1.254 |
| Stoke | 3.649 | 4.781 | 11,186 | 454 | 4.059 | 55.717 | −0.191 | 9.644 | 27.418 | −2.234 |
| Glasgow | 3.929 | 4.545 | 5,720 | 235 | 4.108 | 55.277 | −0.205 | 8.871 | 27.2 | −0.528 |
| Stockport (pilot) | 2.465 | 5.568 | *Removed* | | | | | | | |

The Prevalence column records the age-standardized CAD prevalence rates (as percentages) in 2010, while the Incidence column records CAD incidence rates (as percentages) from 2010 to 2021, calculated based on all new CAD events during that period. Apart from the first 2 columns, the remaining columns refer to the summary statistics for each UKB assessment center from the data set analyzed in this paper. Numbers in italics highlight the most extreme values in that column. Note that Stockport was a pilot center and was not analyzed and Swansea and Wrexham were mobile assessment units. The CAD incidence rates for each center in the complete data set are different from the corresponding incidence rates shown in the third column, because UKB participants who were free of CAD or who reported as having a incidence CAD event but did not have complete information for PCE score calculation or CAD-PRS, were not included in the data set analyzed.

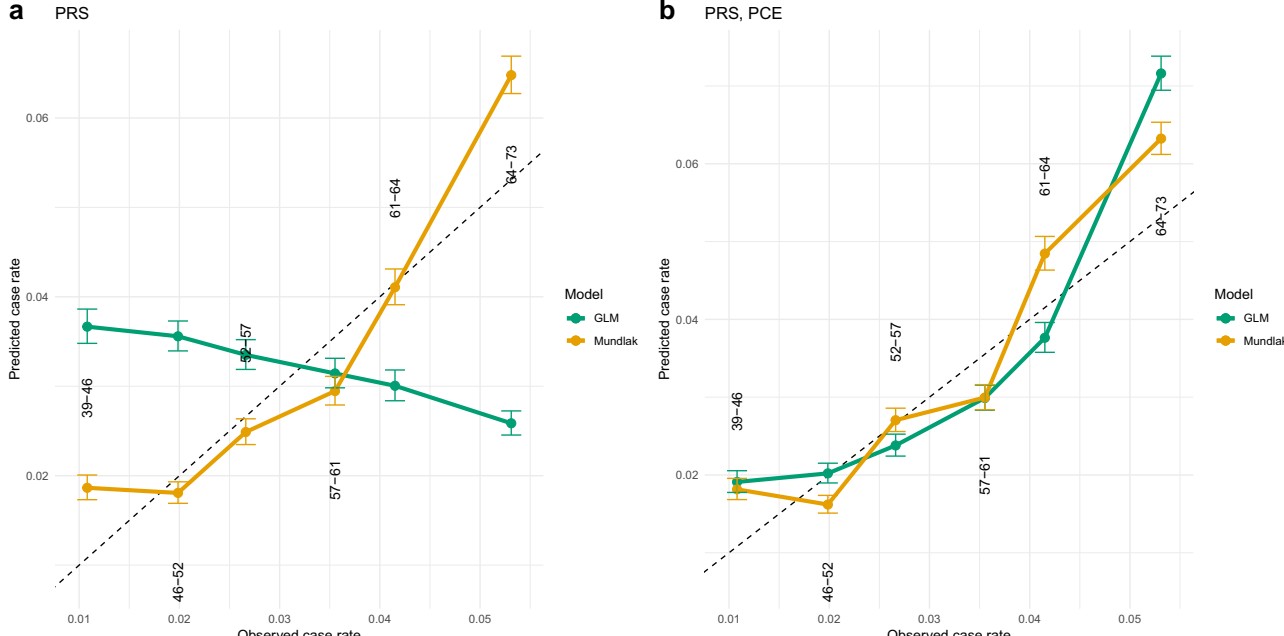

**Fig. A1.** LOCOCV predicted case rates from GLMs regressed on a) PRS only and b) PRS and PCE. Observed case rates are CAD incidence rates for each age group and predicated case rates are the mean of the predicted rates for all participants in the same age group.

**Table A2.** The AUC for each Mundlak model trained and tested on the same complete data set.

| Covariates | GLM AUC | Mundlak GLM AUC |
|---|---|---|
| PCE | 0.7303 | 0.7318 |
| PRS | 0.6318 | 0.6874 |
| PRS + PCE | 0.7515 | 0.7526 |
| PRS + PCE + BMI | 0.7523 | 0.7534 |
| PRS + PCE + TDI | 0.7523 | 0.7534 |
| PRS + PCE + BMI + TDI | 0.7532 | 0.7541 |

PCE, pooled cohort equation; PRS, polygenic risk score; BMI, body mass index; TDI, Townsend deprivation index.

**Table A3.** Age groups with proportions.

| Age group | 2013 ESP % | Adjusted 2013 ESP % |
|---|---|---|
| [35,39] | 0.070 | 0.137 |
| [40,44] | 0.070 | 0.137 |
| [45,49] | 0.070 | 0.137 |
| [50,54] | 0.070 | 0.137 |
| [55,59] | 0.065 | 0.127 |
| [60,64] | 0.060 | 0.118 |
| [65,69] | 0.055 | 0.108 |
| [70,74] | 0.050 | 0.099 |

ESP, European standard populations. Figures in the 2013 ESP % column are published proportions for each age group from the 2013 ESP distribution. Figures in the adjusted 2013 ESP % column are adjusted from the 2013 ESP % column so that the sum of the proportions in the study equals 1

and longitude information, and then used these to create CAD rate maps.

## Age-standardized rates

UKB participants were enrolled between the ages of 37 and 73. To generate age-standardized CAD prevalence rates, we first converted the original proportions for 8 age groups from the 2013 European standard populations distributions into adjusted proportions to make the total proportion equal to 1. The age-standardized prevalence for each assessment center is calculated as the sum of the adjusted prevalence from each age group. The adjusted prevalence is the original prevalence for each age group multiplied by the corresponding adjusted 2013 ESP proportions. Table A3 shows the age groups and the corresponding proportions.

## Influence of weights on within-group correlation in structured permutations

The sample correlation and 95% CI of PRS and PCE across all individuals is 0.00926(0.005, 0.0131). Using our method for group/center label permutation wherein group sizes are preserved and group rates are approximately preserved from the unpermuted data, we observed the following results. When $\alpha_1 = \alpha_2 = 0.2$, the within-group correlation for the min(y) group is $-0.0028$ $(-0.062, 0.056)$ and for max(y) it is $-0.044(-0.0699, -0.018)$. When $\alpha_1 = -0.5, \alpha_2 = 0.5$, the within-group correlation for the min(y) group is $0.055(-0.0038, 0.114)$ and for max(y) it is $0.295(0.271, 0.318)$. So the max(y) group individuals have *positive* correlation between PRS and PCE, as the individuals have an expectation of low values of PRS and high PCE in this case. The max(y) group individuals have positive correlation between PRS and PCE as the individuals with high $\mathbb{E}[y]$ have low values of PRS and high PCE, based on Equation 2. Interestingly, there is a positive linear relationship between the correlation and the rank of the group: as we go from lower ranked groups based on values of y to higher ones, we see increasing within-group correlation. This achieves the effect of negative group-wise correlation but slightly positive individual-level correlation overall. This linear relationship is negative in the case when $\alpha = (0.2, 0.2)$. Figure A2 plots within-group correlation between PRS and PCE for the two structured permutation settings against the group-specific CAD rate.

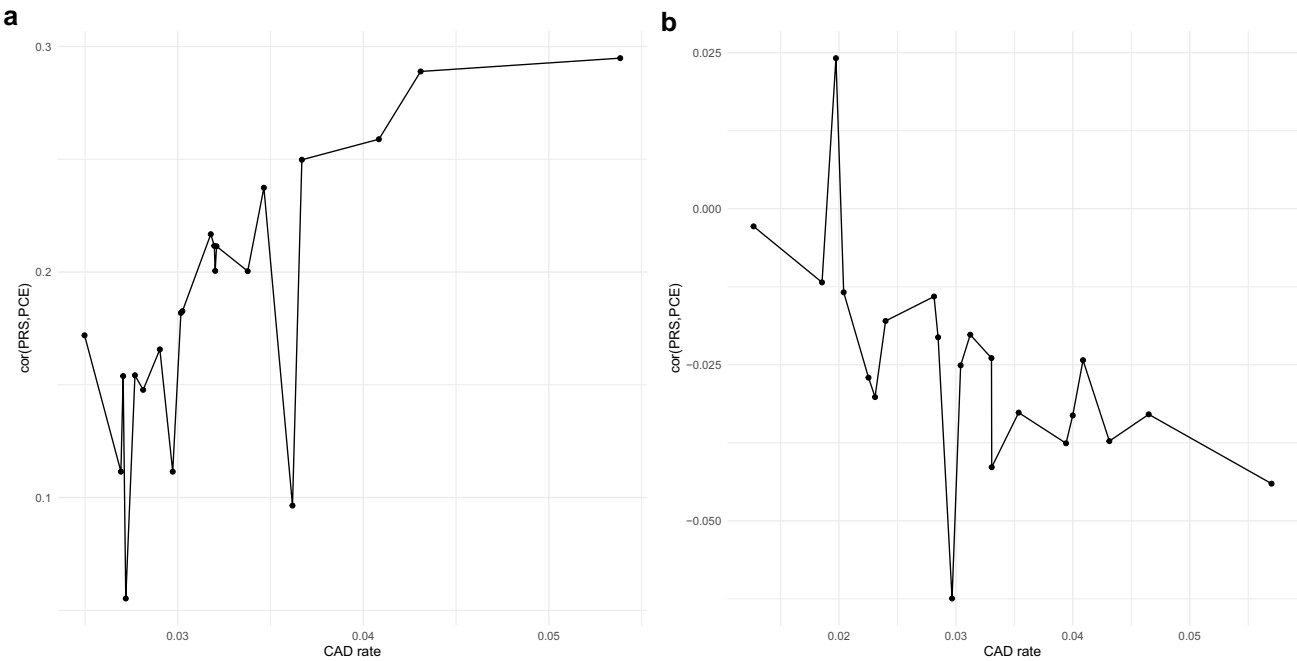

**Fig. A2.** Within-group correlation between PRS and PCE plotted against group-specific case rates for the permuted groups, a) when $\alpha_1$ and $\alpha_2$ from Equation 2 have the same sign (both positive) and b) when $\alpha_1$ and $\alpha_2$ from Equation 2 have opposite signs.