## [Peer Review File · Genetics]

How group structure impacts the numbers at risk for coronary artery disease: polygenic risk scores and non-genetic risk factors in the UK Biobank cohort

Jinbo Zhao, Adrian O'Hagan, and Michael Salter-Townshend

NOTE: The reviews and decision letters are unedited and appear as submitted by the reviewers.

In extremely rare instances and as determined by a Senior Editor or the EIC, portions of a review may be redacted. If a review is signed, the reviewer has agreed to no longer remain anonymous.

The review history appears in chronological order.

Review Timeline:

Submission Date:	2023-07-12
Editorial Decision:	2023-09-01
Resubmission Received:	2024-03-22
Accepted:	2024-05-08

September 1, 2023

GENETICS-2023-306331

How group structure impacts the numbers at risk for coronary artery disease: polygenic risk scores and non-genetic risk factors in the UK Biobank cohort

Dear Dr. Salter-Townshend:

Three experts in the field have reviewed your manuscript, and I have read it as well. There was general enthusiasm for exploring the use of the Mundlak model in order to handle structure when estimating group level risks. While your manuscript is not currently acceptable for publication in GENETICS, we would welcome a substantially revised manuscript. Both reviewers have comments and concerns to be addressed in a revised manuscript. You can read their reviews at the end of this email.

The reviewers concerns largely focused around clarity of presentation and addition of statistical tests to establish the significance of results. Additional text explaining the public health utility of computing these estimates is also requested. In addition to the reviewer's comments I request that you add content related to population structure. It has been repeatedly shown that PRS, even within fine-scale populations in the UK, can have different means and accuracies between populations independent of actual genetic load (see Martin et al AJHG 2017 and subsequent work). How would such differences impact the analyses conducted in this paper? We look forward to receiving your revised manuscript. Please let the editorial office know approximately how long you expect to need for revisions.

Upon resubmission, please include:

1. A clean version of your manuscript;
2. A marked version of your manuscript in which you highlight significant revisions carried out in response to the major points raised by the editor/reviewers (track changes is acceptable if preferred);
3. A detailed response to the editor's/reviewers' feedback and to the concerns listed above. Please reference line numbers in this response to aid the editor and reviewers.

Your paper will likely be sent back out for review.

Additionally, please ensure that your resubmission is formatted for GENETICS
<https://academic.oup.com/genetics/pages/general-instructions>

Follow this link to submit the revised manuscript: Link Not Available

Sincerely,

Noah Zaitlen
Associate Editor
GENETICS

Approved by:
Sharon Browning
Senior Editor
GENETICS

Reviewer #1 (Comments for the Authors (Required)):

In the study by Zhao and colleagues, the team pinpointed geographical variations in CAD age-standardized rates across UKB assessment centres. They also observed distributional disparities in PRS and PCE risk between UKB participants from England and Scotland. To address these findings, they introduced a Mundlak model that corrects for systematic biases at the cohort level, offering a means to predict the number at risk in new groups using individual-level data samples.

After reviewing the paper, I find the methods and results to be fundamentally robust. My comments will focus on potential enhancements to the manuscript and specific clarifications that the authors might consider adding.

Comments:

1) Manuscript could benefit from a more concise presentation without compromising the depth of information. Here are some recommendations:

Introduction: The section on CAD can be streamlined to focus on the most pertinent details.

Methods: Certain aspects, such as the workings of the permutation test or the intricacies of the Mundlak model, might be better suited for the appendix.

Figures & Tables: With 16 figures presented, consider amalgamating relevant ones into multi-panel visuals. Moreover, some tables might be more appropriate for supplementary material (especially since most are plotted anyways).

Ultimately, these suggestions aim to enhance the paper's readability. The final decision rests with the authors and the editorial team but refining the presentation would help readability.

2) Beyond noting significant distributional differences in PCE risk through the Mann-Whitney p-value, did the authors use additional measures (such as Cohen's d or Cliff's Delta) to quantify the difference? While the Mann-Whitney U test discerns whether one group generally has higher values than the other, a significant result only highlights a distributional difference without elucidating the magnitude of this difference, a concern, particularly in studies with extensive sample sizes.

3) I was unable to locate specifics about the number of permutations conducted. Please include this information if it hasn't been provided. I presume the authors utilized stratified permutation to maintain group integrity, permuting only the group labels. Can you confirm this?

4) While significant geographical variations were noted, the biological implications of these findings remain less explored. What is about Wrexham that makes it similar to others?

5) In Table 9, the correlation of observed and predicted group rates for LOOCV GLM is -0.053 when considering PRS + Age. Could you clarify why this is the case? The result appears counterintuitive.

6) The majority of comparisons involving the Mundlak model are juxtaposed against a basic GLM. I'm curious how its performance would fare when contrasted with a latent variable or mixed model that incorporates group-specific intercept terms. Additionally, LOOCV can potentially yield an optimistic performance estimate, particularly for intricate models. Have the authors validated the model's performance using alternative cross-validation methodologies?

Reviewer #2 (Comments for the Authors (Required)):

Looking at disease incidence or other summaries for subgroups within a study data set is of general interest. The authors of this manuscript are specifically interested in predicting the risk of CAD at the regional level and comparing regions. They use subgroups based on assessment centre, which is a natural grouping for analyses of UKBB data. They characterize centre-specific CAD incidence using group mean predicted incidence from disease risk models that leverage polygenic risk scores (PRS) and other disease-related risk factors, including a non-genetic or family history based combined risk score (PCE).

Averaging results from standard risk prediction models (GLM) across study participants at each centre gave group average predicted CAD incidences that were poorly correlated with observed group/centre incidence rates. They suggest that the issue is structure in the data set leads to different joint distribution of PRS and risk factors at different centers. With simulations they illustrate a sorting process whereby study participants in different slices of the risk-factor joint distribution are selected by different assessment centres. This selection process creates structure that foils their baseline approach for estimation and comparison of group-level mean disease incidence. However, a less-well-known model for group-level analyses they call the Mundlak model does allow group-level predictions of mean CAD incidence that are better correlated with observed mean incidence. They apply the Mundlak model to predict mean centre-specific CAD incidence with results better correlated to the observed group mean incidences.

The Mundlak model is an established statistical tool that has not seen use in genetics studies. The analysis of predicted CAD incidence by recruitment centre is an appropriate application of the approach. There is much interest in applying PRS and other risk-scores to predict and characterize disease incidence in biobanks and patient populations. The improved performance for group mean disease incidence prediction demonstrated here for the leave-one-centre-out cross-validation results when using the Mundlak model suggest that this approach can give more generalizable prediction results than standard alternatives. In applications, a Mundlak model using natural groupings such as nation, county, or centre for UKBB could be fit, and then predictions for new or individuals could be made that take the new individuals group info into account if known, but could be made robustly if the group is not known or was not used in the model-training (suggested by the COCOV).

This is a useful contribution to the genetics literature and should be of interest to genetics and epidemiology researchers.

Some issues and suggestions:

Page 4, line 70

This sentence doesn't make sense. Cut it and put the citation (Hart, 2001) on the previous sentence.

"The Mann-Whitney test is considered to be a test of population medians and is accompanied by equally important differences in shape, but the Mann-Whitney test cannot discern differences between two groups with the same median, but can discern different variances or shapes, as this test analyzes only the ranks"

page 5, line 53

Condition on PRS=prs: $\Pr(L>T|PRS=prs) = \Pr(L-prsj > T - prsj | PRS=prs)$

prs isn't just some constant, it is the value of the random variable PRS

page 6, line 62

Maybe just call alphas weights? Correlation with what? α_1 is related to covariance of Y and PRS, and $\text{var}(Y)$ is defined as 1, but are PRS, PCE scaled for $\text{cor}(PRS, Y)$ to be α_1 ? PRS was scaled page 5 line 57 to match the complete study CAD incidence rate, right?

It is confusing to call the alphas correlations, because a key phenomenon from ascertainment or selection is induction or modifications of correlations between variables related to the selection process (PRS and PCE) by selection on y.

Suggested additions:

For the simulations of groups with sorting/assignment by $y = \alpha_1 * PRS + \alpha_2 * PCE + \text{noise}$, it would be interesting to see the global correlation of PRS and PCE, and then the correlations of PRS and PCE within groups, at least for the $\max(y)$ and $\min(y)$ groups. Selection by y alters the correlation between variables associated with selection (PRS and PCE here). I think this difference in correlation between variables is more important than the differences in the group mean values for the variables across groups, which is the emphasis in this paragraph. I would suggest looking at these within-group correlations for $\max(y)$ and $\min(y)$ groups, summarizing results in a table in the appendix, and discussing the takeaway with the other simulation results on page 14.

Page 12, line 18.

Clarification. I assume these LOCOCV predictions are made by fitting the Mundlak model on observations from a set of groups/centres, and then for a prediction into a new-unknown group, the unknown group means are assumed to be zero. Is that correct? Maybe make that explicit here, or correct me if I'm wrong and write the correct details here.

From tables 8 and 9, LOCOCV group correlation (predicted incidence, observed incidence) is stronger for Mundlak GLM than GLM. Beyond group analyses, an application would be more robust prediction models, regardless of whether group is known or if there has to be an out-of-distribution prediction for someone from a new group. Towards that application, it would be interesting to see another table with AUCs for a set of models evaluated in a complete data set (not lococv), with three columns for: GLM, Mundlak GLM, Mundlac GLM with group means set to zero for predictions.

Figures:

Figure 1. This isn't a proper flow chart. Branches should be disjoint options as the top three boxes in the middle (prevalent CAD, incident CAD, no CAD), but the 4th and 5th boxes and the box on the right (complete covariates) are sequential. Make [Total (left box)] -> [prevalent, incident, no CAD (3 boxes)] -> incident and no-cad flow forward and split on Y/N have PRS -> yes PRS move forward on Y/N PCE score -> Yes PCE move forward on Y/N other covariates -> Y move to study as usable incident and no-CAD participants

Figure 2. Distracting that 2a is a different format than the (b) and(c). Maybe move (a) to its own figure earlier in the paper when it's mentioned that the focus of the paper is on relative mean prediction performance in various substrata of the UKB data set, and the particular groups of interest will be the recruitment centres.

Figure 3. In the caption, mention that the dotted lines are for group means (medians?)

Figure 8. Some plotting error in panel (b) a lot of lines are missing. With this choice of y-axis range, predicted rates for Barts would be off-scale

Figures 9,10,17. These are correlations, not p-values. Fix labels on color scale and captions.

Figures 9, 10. Except for PCE, everything in Figure 9 is also in Figure 10. Combine these.

Figure 10. Note the sign flips in (a) vs (b) for $\text{cor}(PRS, HDL)$ and $\text{cor}(\text{systolic blood pressure}, TDI)$ by outlining those boxes in the plots, and in the caption.

Figure 20. Why highlight the model with PRS alone? Figure 12 (the other prediction interval figure) highlights PRS + PCE, and by Table 10 PRS+PCE has better performance for the age stratified data. Probably should highlight PRS+PCE results.

Tables:

Table 1. The row and column labels are "AUC \ CAD definition". Should be "PRS model \ CAD definition"

Table 6. AIC, BIC, LL, Deviance aren't informative unless you're comparing models or model fits, drop these. In model specification (which variables used), generally "PCE" is used (say in Table 5). Why is it "PCE risk" in Table 6? Are the variables scaled in a way for the regression coefficients to have interpretable units? What are the units?

Table 10. For more consistency with other tables, update column names to ("Variables", "GLM AUC" and "Mundlak GLM AUC")

Reviewer #3 (Comments for the Authors (Required)):

Review of GENETICS-2023-306331

Summary

This study investigates how coronary artery disease (CAD) can be predicted from genetic and non-genetic variables obtained for UK Biobank (UKB) participants. The main focus is at predicting risk at group level i.e. predicting the fraction of individuals in a particular assessment center that will get a diagnosis.

The authors find that in making such predictions it appears beneficial to include group mean effect (Mundlak model). They find that the polygenic risk score (PRS) seems to have a groups level effect.

Strength and weaknesses

The study is based on a very large cohort from the UKB with genetic and clinical risk estimated available more than 250,000 individual with more than ten years of follow-up. Obtaining accurate estimates of risk of CAD is of obvious clinical relevance as pharmacological prevention is readily available, and important questions in the area is how PRS interplay with non-genetic risk factors and how ascertainment effects and geographical biases may impact on such risk estimates. However, the specific aim of this study is "predicting the risk of CAD at the regional/country level", meaning the authors are interested in predicting the fraction of individuals in a UKB assessment center that will develop CAD, more than predicting who will develop CAD, which I find a bit obscure. Further, many of the central points highlighted as main finding could in my reading only be trends in the data as all estimates lack confidence intervals, and no statistical tests are applied to quantify the statistical significance of the findings.

Overall assessments

While I fill that the general topic is of broad interest, this specific study takes a somewhat unusual angle on the topic, and no convincing explanation why this approach is useful is provided (See Major Comments 9a-b). The general lack of confidence intervals, p-values or other means of justification that the trends described are anything but sampling variance unfortunately means, that I do not find that the results presented provide strong support for the conclusions reached.

Major Comments

1. I find it quite hard to follow the study flow chart in Figure 1. Some boxes seem to contain individuals that were excluded while other seem to be different subset of individuals that were included, but it's unclear for me how to read the diagram. My understanding from the text is that the 263,087 individuals were selected based on: (a) being in the UKB, (b) being classified as white British, (c) not having CAD before joining UKB, (d) having an available CAD-PRS, (e) not having an ASCVD event before joining UKB, (f) LDL-C < 190, (g) not having extreme blood pressure or cholesterol, and (h) not having missing values for PCE variables.

I think a more traditional stepwise study flow diagram could improve the transparency of the study. For instance, one showing how many individuals were excluded by each of the steps (a-h) and how many remained.

2. The first result of the manuscript is that there is "geographical variations in age-standardized rates of CAD between assessment centres". Here the authors describe how they calculated a weighted sum of the fraction of individuals with prevalent CAD in each of eight age-strata with weights corresponding to the European standard population distribution.

a. I'm struggling to find the actual estimates. I see some colored dots in Figure 2b, but the unit is not given anywhere (are these percentages?).

b. Additionally, no confidence intervals are provided, and no statistical test is applied to demonstrate that the age-standardized prevalence and incidence differ more across the assessment centers than would be expected by change (i.e. due to measurement error). Without such a test, I find it hard to judge whether the claim of "geographical variations in age-standardized rates of CAD between assessment centres" is supported by the data.

c. would it be relevant to standardize the prevalence rates by other factors than age (e.g. sex)?

3. A second result is that both the PGS and PCE seem to differ between assessment centers.

a. While we do get many technical details about the calculation of the LDpred2 PRS (which wasn't used for the main analyses anyway) we get very few for the UKB-CAD-PRS. I'm guessing that the PRS being used is this one

(<https://biobank.ndph.ox.ac.uk/ukb/field.cgi?id=26227>) and while I can find some information in Supplementary Tables of Thompson et al, I think this information should be summarized in the methods section (e.g. what was the training sample of PGS, what was the sample size, phenotype definition etc.)

b. Should there be some multiple testing correction given the large number of tests (22x22) presented in Figures 4, 5 and 6? I agree that there seems to be more significant ($p < 0.05$) differences than we would expect by chance, but I think a formal test would be appropriate.

4. A third result is that the authors try to predict CAD using a set of risk factors including PGS, PCE, BMI and TDI. AUC values for are provided in Table 4, but without confidence intervals. In particular, I find that the correlation coefficients also provided in Table 4 are hard to interpret without a confidence interval since if I understand it correctly, they are based on only 22 observations.

5. With reference to Figure 7, the authors describe the correlation between observed and predicted case rates at the different assessment centers as low. To show that this correlation is lower than expected, the authors use a rather untraditional approach in which they create groups with different rates of cases and controls. They show that in this random grouping, the correlation between the observed case rates and predicted case rates appears higher. However, I think this analysis should be extended in several ways:

a. How many random groups were there? although it says nine groups in the text, I see 15 dots in Figures 8a. Why not 21 as in the actual data?

b. Why were similarly sized groups picked here? It looks like the outlying groups in the real data tend to be the smaller assessment centers, which we would also expect to have noisier estimates, right?

c. Was any statistical test used to quantify the difference between Figures 7 and Figure 8a, or just visual inspection?

d. Was the random grouping just done once? Why not do it 10,000 times to get an empirical distribution of the correlation coefficient under the null?

I think all of these questions above could be addressed by a more traditional permutation test, where samples are permuted within cases and within controls.

6. In Figures 9-10 the authors again make a comparison between correlation coefficients estimated based on >280,000 data points and 21 datapoints given that the correlation coefficient has a standard error $= \sqrt{(1-r^2)/n}$, I'm unsure which of the group-level correlation coefficients should be considered significant. I have a hard time following the interpretation of these results.

7. The section at page 10 lines 43-60, seems a bit speculative and without a simulation experiment or a DAG analysis to back it up, I think it shouldn't be included as a result in this paper.

8. The main result of the paper is to fit a Mundlak model, where in addition to the individual PGS and PCE, two new variables (group mean PRS and group mean PCE) are created which contain the per assessment-center average value of the PRS and PCE, respectively. In Table 6 we see that there seems to be a significant effect of group mean PRS but not of group mean PCE. However, comparing the AUC in Table 4, i.e. without the Mundlak effect (AUC-PRS=0.6318) to that in Table 5, i.e. with the Mundlak effect (AUC-PRS=0.635) the effect seems to account for rather little variability in CAD risk. Looking closer at Table 6, it seems that the beta-value for the group mean PRS is much higher than that for the PRS. My intuition is that this difference indicates that the PRS is somehow capturing other effects than causal genetic effects. Do you agree?

9. The authors state that the correlation as the assessment center level is higher for the Mundlak model than the simple GLM.

a. First of all, I am not really able to think of a situation in which you would want to make a prediction at the assessment center level. Statins, which the authors mention as a motivation for predicting CAD, are usually prescribed at individual level, not at assessment center level. If the authors have a particular use of such predictions in mind, I would make it more explicit.

b. In the last sentence of the Conclusions paragraph "screening programmes and early interventions" are mentioned. If the authors are aware of any initiatives to implement such things at UKB assessment center level, and if that was the real motivation for running the study, I would suggest pointing it out early in the introduction.

c. I agree with the authors that the cross-validated model is a good choice when evaluating the correlation as the assessment center level. Bearing in mind that you have only 22 assessment centers and that you are fitting models in which the explanatory variables are constructed by clustering at assessment center level overfitting seems likely. Indeed, all correlations presented in Table 5 and Table 6 are higher than the cross-validated Mundlak model version presented in Table 8 and Table 9. If the authors agree that the correlations in Table 5 and Table 6 and the corresponding figures (Figure 11, 12 and 13) are overfitted is there any reason to include them in the paper?

d. Given that Table 5 and Table 6 are likely overfit Tables 8 and 9 seem to be the right place to judge if the Mundlak model is effective. However, without confidence intervals of the correlation coefficients it is difficult to judge whether the estimated correlations are significantly higher.

-
Minor comments

Figure two why does the legend have same label in b) and c)

Figures 9, 10, 17 says "p-values", but I assume these are correlation coefficients. I would like to see the actual p-values, though.

Table 2: "4. Randomly (...) then move the same number of controls from group B to group A." should be "from group D to group C", right?

Page 4 line 31:

"Of the 502,401 UKB participants, 35,308 were identified as participants with a first ASCVD epidemic event." I don't know what epidemic refers to here, but it sounds misplaced.

Table S6: "bold italics are the highest in that column" -0.287 seems to be the lowest?

Associate Editor Comments:

Response Letter for “How group structure impacts the numbers at risk for coronary artery disease: polygenic risk scores and non-genetic risk factors in the UK Biobank cohort”

Jinbo Zhao, Adrian O’Hagan, Michael Salter-Townshend

Dear Associate Editor of GENETICS,

Many thanks for the in-depth reviews provided by you and the three reviewers of our manuscript. The criticisms were uniformly constructive and clear. We hope that you agree that our incorporation of them in our paper has greatly strengthened our contribution. Changes in the revised manuscript have been highlighted in red for clarity. We also provide a point-by-point response to all queries and comments raised below.

1 Decision letter from the editor

Three experts in the field have reviewed your manuscript, and I have read it as well. There was general enthusiasm for exploring the use of the Mundlak model in order to handle structure when estimating group level risks.

Many thanks for agreeing that our proposed model has merit.

The reviewers concerns largely focused around clarity of presentation and addition of statistical tests to establish the significance of results.

We have rewritten the sections where inadequate clarity was highlighted by reviewers and we have provided additional statistical tests and measures of uncertainty for each request as detailed below.

Additional text explaining the public health utility of computing these estimates is also requested.

We agree that this was under-motivated in the original version and have included additional references and text in the “Study aim” section. We have added text and additional references to other studies investigating the ability of PRS modelling to estimate rates of disease across sub-populations. The existence of this literature shows that there is an interest and that current models underperform relative to expectations.

1.1 Population Structure

In addition to the reviewer’s comments I request that you add content related to population structure. It has been repeatedly shown that PRS, even within fine-scale populations in the UK, can have different means and accuracies between populations independent of actual genetic load (see Martin et al AJHG 2017 and subsequent work). How would such differences impact the analyses conducted in this paper?

Two very recent papers investigate the causality and mechanism of population structure (or ancestry) on Polygenic Risk Score accuracy variation: [Hou et al., 2023] and

[Hu et al., 2023]. In short, both show that underlying causal effect sizes in GWAS are actually very similar (or even the same) in different ancestries. They show this by examining admixed individuals and including (estimated) local ancestry in the GWAS model, leading to estimates of causal SNP effects that are highly consistent across differing ancestries (e.g. European and African).

However, they arrive at differing conclusions: [Hou et al., 2023] conclude that the differences observed across populations are due to gene-environment interactions, because they control for environment and then get similar effect size estimates. In contrast [Hu et al., 2023] don't control for environment, but do include local (specific to chromosome segment) ancestry of each individual ([Hou et al., 2023] include genome-wide averages only). They still get consistent estimates of effect sizes across populations, so they conclude that there is no cross-ancestry difference of either gene-gene or gene-environment interactions. They note that gene-environment interactions do impact prediction accuracy, but acting in similar ways across different populations. In other words, the interactions work the same, but differing environments leads to different contributions to risk via gene-environment.

We have now included references to both studies, plus a summary based on the above in the section on Geographical variations in cardiovascular disease prevalence across the UK. In the "Conclusion" section we now also quote the [Abdellaoui et al., 2019] finding that "many of the regional differences in health-related outcomes are not likely to be caused by differences in genetic variants that are directly causal for the health outcomes themselves, and are more likely to be due to environmental factors correlated with regional SES and therefore also with the SES-related genes of inhabitants." In other words, for these socioeconomic status phenotypes, our Mundlak model would work well precisely because it models this correlation between gene and environment. Our updated manuscript demonstrates this ability for CAD and we now reference the similarity between our model and one of those assessed in [Abdellaoui et al., 2022] in which it is shown that the inclusion of a regional-average PRS term improves SES phenotype prediction in a siblings based study using UK Biobank data.

Finally, we cite [Lin et al., 2023] who showed that "only up to three PCs appears to be sufficient for controlling population stratification for most outcomes, whereas including other covariates (particularly age and sex) appears to be more essential for model performance."

1.2 Summary of improvements in the revised version

Thank you for a thorough review, which has enabled us to greatly improve our contribution. In particular, we feel that Reviewer 2 provided a succinct summary of our study. Reviewers 1 and 3 provided valid and valuable constructive criticism around better motivating for our work. We have therefore expanded upon the motivation and validation of our approach in particular. Below we summarise the main changes we have made in light of the reviews, followed by a reply to each reviewer comment in turn.

Our focus is on demonstrating the usefulness of the Mundlak model for dealing with structure in estimating group level risks and identifying the conditions under which the Mundlak GLM outperforms the standard GLM for predicting the CAD rate within a group, based on covariate information along with training on other groups. The clarity of our central message is much improved due to the review process as we have reduced the overall number of tables and figures, while now including the suggested additions:

- Formal statistical tests to identify regional differences among PRS and PCE and

other pairs of covariates.

- Inclusion of confidence intervals in plots and tables to support the increased accuracy attributable to the Mundlak GLM.
- Refined the simulation / permutation experiments to help clarify the conditions under which the Mundlak model will be useful and why.
- We have added motivation and greater explanation to the Leave-one-centre-out cross-validation method for assessing performance and moved the section to before the results. As suggested, we now present results using this alone rather than in-sample prediction performance.
- Simplified plots by restricting to a common format of one GLM and the corresponding Mundlak GLM for each figure, with confidence intervals / error bars. We also adopt a consistent colour scheme.
- We have combined various tables as suggested by the reviewers.
- Removal of the case-control swaps simulation results and a re-write of the group label permutation experiment and results. Considering the constructive feedback from the reviewers, we feel this simplifies and strengthens our message. We have also adapted the group permutation experiment so that group sizes and approximate rates are conserved, relative to the un-permuted dataset. Thus it represents a closer set of conditions to the motivating problem.
- More detail and references underpinning how our approach relates to, but is different from, controlling for population structure via regression on e.g. Principal Components.

2 Comments from the first reviewer

In the study by Zhao and colleagues, the team pinpointed geographical variations in CAD age-standardized rates across UKB assessment centres. They also observed distributional disparities in PRS and PCE risk between UKB participants from England and Scotland. To address these findings, they introduced a Mundlak model that corrects for systematic biases at the cohort level, offering a means to predict the number at risk in new groups using individual-level data samples.

After reviewing the paper, I find the methods and results to be fundamentally robust. My comments will focus on potential enhancements to the manuscript and specific clarifications that the authors might consider adding.

Many thanks for your encouragement. We hope that our responses to your comments below further enhance our contribution.

1. Manuscript could benefit from a more concise presentation without compromising the depth of information. Here are some recommendations:
 - (a) Introduction: The section on CAD can be streamlined to focus on the most pertinent details.
We have shortened and edited the section as suggested.

- (b) Methods: Certain aspects, such as the workings of the permutation test or the intricacies of the Mundlak model, might be better suited for the appendix. Agreed, we have removed the section on case-control swaps and refocused the section "How the Mundlak Model Works". We feel that this latter section is crucial to the paper and have therefore retained it in the main text.
 - (c) Figures & Tables: With 16 figures presented, consider amalgamating relevant ones into multi-panel visuals. Moreover, some tables might be more appropriate for supplementary material (especially since most are plotted anyways). Thanks for highlighting this. Many figures are now amalgamated, and tables merged and moved to the first Appendix.
 - (d) Ultimately, these suggestions aim to enhance the paper's readability. The final decision rests with the authors and the editorial team but refining the presentation would help readability.
Agreed and we hope you find the revised paper more readable.
2. Beyond noting significant distributional differences in PCE risk through the Mann-Whitney p-value, did the authors use additional measures (such as Cohen's d or Cliff's Delta) to quantify the difference? While the Mann-Whitney U test discerns whether one group generally has higher values than the other, a significant result only highlights a distributional difference without elucidating the magnitude of this difference, a concern, particularly in studies with extensive sample sizes.
- (a) We did also examine Cohen's d for PCE and PRS in the complete data. However, we did not find a significant difference in effect size between almost all pairs of centres or between regions. The only significant differences in effect size are for PCE risk between Barts and Stockport or Swansea under the Cohen's d test (estimated effect size statistic is between 0.2 and 0.5). We have added these results to the section "Statistical Tests". Cohen's d compares the effect sizes and uses the difference between means divided by the pooled standard deviation. However, as with [Yang et al., 2021] we believe it is focussed on the wrong thing. Two populations can have no difference in mean PRS, but have very different rates as the upper tails are where cases occur. In our study, we aim to explore the effectiveness of the Mundlak model for group structure within the UKB participants, which has been shown to have geographical clustering of health-related outcomes [Abdellaoui et al., 2019]. The use of the Mundlak model requires the difference between the group means, as this model adds the group-level mean of each of the included explanatory variables. We have added text to this effect to our manuscript. Cliff's Delta is for ordinal data and we do not explore this further.
 - (b) We now also include other tests to quantify the difference. For example, to correct for the familywise error rate, as advised by the third reviewer in comment 3(b), we used Benjamini & Hochberg (BH) corrected p-values from permutation tests ([Camargo et al., 2008]). See Figure 1 below (Figure 6 in the revised paper) for PCE risk and Figure 2 below (Figure 5 in the revised paper) for UKB CAD-PRS.
3. I was unable to locate specifics about the number of permutations conducted. Please include this information if it hasn't been provided. I presume the authors utilized stratified permutation to maintain group integrity, permuting only the group labels.

Figure 1: P-values from permutation tests after Benjamini & Hochberg corrections for PCE risk.

Can you confirm this? Apologies for the omission. We now report in the paper that there were 2000 permutations conducted. We also added the BH corrected p-values for the permutation tests, as suggested.

4. While significant geographical variations were noted, the biological implications of these findings remain less explored. What is it about Wrexham that makes it similar to others? We assume you mean to ask what makes Wrexham different to the others: Compared to other assessment centres, Wrexham has the smallest number of participants (only 304) and the lowest mean PRS score, but the highest mean BMI value (new Table 4). This is now clarified and also indicated by the wider confidence intervals in Figures 8 to 12 of the revised paper.
5. In Table 9, the correlation of observed and predicted group rates for LOCOCV GLM is -0.053 when considering PRS + Age. Could you clarify why this is the case? The result appears counterintuitive. The Pearson correlation between Age and PRS is -0.04 for the complete data set and 0.1 for the mean values between UKB assessment centres. The Pearson correlation between Age and CAD rate is -0.17 (as per the covariance heatmap). Hence this is a really good example of Simpson's paradox at work and an instance where the use of the Mundlak model leads to more accurate group-wise predictions. Although both age and PRS are positively correlated with

Figure 2: P-values from permutation tests after Benjamini & Hochberg corrections for UKB CAD-PRS.

risk, they are negatively correlated with each other. We have added a note to this effect to the text in the Section Results from standard GLMs.

6. The majority of comparisons involving the Mundlak model are juxtaposed against a basic GLM. I'm curious how its performance would fare when contrasted with a latent variable or mixed model that incorporates group-specific intercept terms. Additionally, LOCOCV can potentially yield an optimistic performance estimate, particularly for intricate models. Have the authors validated the model's performance using alternative cross-validation methodologies?

This study aims to predict the group risk for a new group. The Mundlak model does not require an estimate of the group effect (calculated using observed rates of a sample from the group), whereas the latent variable or mixed model requires this group effect estimate to make a prediction. Thus our proposed LOCOCV aligns with the intended purpose; to fit a model on a set of groups where both sample rates and covariates are available and then to use this fitted model to predict rates in another group for which only the covariates (samples of PRS, PCE, etc) are available. We have clarified this further in the "Study aims" section.

3 Comments from the second reviewer

Looking at disease incidence or other summaries for subgroups within a study data set is of general interest. The authors of this manuscript are specifically interested in predicting the risk of CAD at the regional level and comparing regions. They use subgroups based on assessment centre, which is a natural grouping for analyses of UKBB data. They characterize centre-specific CAD incidence using group mean predicted incidence from disease risk models that leverage polygenic risk scores (PRS) and other disease-related risk factors, including a non-genetic or family history based combined risk score (PCE).

Averaging results from standard risk prediction models (GLM) across study participants at each centre gave group average predicted CAD incidences that were poorly correlated with observed group/centre incidence rates. They suggest that the issue is structure in the data set leads to different joint distribution of PRS and risk factors at different centres. With simulations they illustrate a sorting process whereby study participants in different slices of the risk-factor joint distribution are selected by different assessment centres. This selection process creates structure that foils their baseline approach for estimation and comparison of group-level mean disease incidence. However, a less-well-known model for group-level analyses they call the Mundlak model does allow group-level predictions of mean CAD incidence that are better correlated with observed mean incidence. They apply the Mundlak model to predict mean centre-specific CAD incidence with results better correlated to the observed group mean incidences.

Thank you for this very clear and accurate summary of our study.

The Mundlak model is an established statistical tool that has not seen use in genetics studies. The analysis of predicted CAD incidence by recruitment centre is an appropriate application of the approach. There is much interest in applying PRS and other risk-scores to predict and characterize disease incidence in biobanks and patient populations. The improved performance for group mean disease incidence prediction demonstrated here for the leave-one-centre-out cross-validation results when using the Mundlak model suggest that this approach can give more generalizable prediction results than standard alternatives. In applications, a Mundlak model using natural groupings such as nation, county, or centre for UKBB could be fit, and then predictions for new or individuals could be made that take the new individuals group info into account if known, but could be made robustly if the group is not known or was not used in the model-training (suggested by the COCOCV).

This is a useful contribution to the genetics literature and should be of interest to genetics and epidemiology researchers.

Many thanks for your encouragement and useful summary. We have incorporated your suggestions as per the below responses.

Some issues and suggestions:

1. Page 4, line 70. This sentence doesn't make sense. Cut it and put the citation (Hart, 2001) on the previous sentence. "The Mann-Whitney test is considered to be a test of population medians and is accompanied by equally important differences in shape, but the Mann-Whitney test cannot discern differences between two groups with the same median, but can discern different variances or shapes, as this test analyzes only the ranks"

Agreed. Thank you for spotting this. We have removed as suggested.

2. page 5, line 53 Condition on $PRS = prsj : Pr(L > T | PRS = prsj) = Pr(L - prsj > T - prsj | PRS = prsj)$, $prsj$ isn't just some constant, it is the value of the random

variable PRS Our apologies, we are unsure of the issue here. $prsj$ is indeed the value taken by random variable PRS for individual j and we are calculating the probability that the total unobserved liability exceeds the threshold given this observed PRS value for individual j . We have edited the text to be more verbose, hopefully this clarifies the issue.

3. page 6, line 62 Maybe just call alphas weights? Correlation with what? α_1 is related to covariance of Y and PRS, and $\text{var}(Y)$ is defined as 1, but are PRS, PCE scaled for $\text{cor}(\text{PRS}, Y)$ to be α_1 ? PRS was scaled page 5 line 57 to match the complete study CAD incidence rate, right? No, $\text{cor}(\text{PRS}, Y)$ is not scaled to be α_1 . Yes, PRS was scaled to the study wide CAD rate.

It is confusing to call the alphas correlations, because a key phenomenon from ascertainment or selection is induction or modifications of correlations between variables related to the selection process (PRS and PCE) by selection on y .

We agree that the term “correlations” is indeed misleading and have therefore changed it to “weights”, as suggested.

4. Suggested additions: For the simulations of groups with sorting/assignment by $y = \alpha_1 * PRS + \alpha_2 * PCE + noise$, it would be interesting to see the global correlation of PRS and PCE, and then the correlations of PRS and PCE within groups, at least for the $\max(y)$ and $\min(y)$ groups. Selection by y alters the correlation between variables associated with selection (PRS and PCE here). I think this difference in correlation between variables is more important than the differences in the group mean values for the variables across groups, which is the emphasis in this paragraph. I would suggest looking at these within-group correlations for $\max(y)$ and $\min(y)$ groups, summarizing results in a table in the appendix, and discussing the takeaway with the other simulation results on page 14.

We have added some text and a new figure as suggested. The global sample correlation and 95% confidence interval of PRS and PCE is 0.00926(0.005, 0.0131). Using our updated method for group / centre label permutation wherein group sizes are preserved and group-rates are approximately preserved from the unpermuted data, we observed the following results: When $\alpha_1 = \alpha_2 = 0.2$ the within-group correlation for the $\min(y)$ group is $-0.0028(-0.062, 0.056)$ and for $\max(y)$ it is $-0.044(-0.0699, -0.018)$. When $\alpha_1 = -0.5, \alpha_2 = 0.5$ the within-group correlation for the $\min(y)$ group is $0.055(-0.0038, 0.114)$ and for $\max(y)$ it is $0.295(0.271, 0.318)$. So the $\max(y)$ group individuals have *positive* correlation between PRS and PCE, as the individuals have an expectation of low values of PRS and high PCE in this case. Interestingly, there is a positive linear relationship between the correlation and the rank of the group: as we go from lower ranked groups based on values of y to higher ones, we see increasing within-group correlation. This achieves the effect of negative group-wise correlation but slightly positive individual-level correlation overall. This linear relationship is negative in the case when $\alpha = (0.2, 0.2)$. We have added Figure 14 in Appendix “Influence of weights on within-group correlation in structured permutations” to demonstrate and discuss this result.

5. Page 12, line 18. Clarification. I assume these LOCOCV predictions are made by fitting the Mundlak model on observations from a set of groups/centres, and then for a prediction into a new-unknown group, the unknown group means are assumed to be zero. Is that correct? Maybe make that explicit here, or correct me if I'm wrong and write the correct details here.

Apologies for the confusion. The LOCOCV prediction on one centre is indeed made by first fitting the Mundlak model on observations from all other centres. As such it represents the closest proxy to our study aim in terms of predicting the group-level rates based on a model fitted to other groups, plus a sample of individuals in the group for whom we have both genetic and non-genetic information. i.e. the PRS and PCE scores from a sample of the population are used to predict the group rate. We have clarified this in the Study Aims section and elsewhere.

6. From tables 8 and 9, LOCOCV group correlation (predicted incidence, observed incidence) is stronger for Mundlak GLM than GLM. Beyond group analyses, an application would be more robust prediction models, regardless of whether group is known or if there has to be an out-of-distribution prediction for someone from a new group. Towards that application, it would be interesting to see another table with AUCs for a set of models evaluated in a complete data set (not lococv), with three columns for: GLM, Mundlak GLM, Mundlak GLM with group means set to zero for predictions.

A new Tables 2 has been created to summarise these and other figures. We have also added text to highlight that the individual level AUC does indeed increase for all models when the Mundlak terms are included. Appendix Table 5 shows AUC for some standard GLMs and corresponding Mundlak GLMs without leaving any observations out. When we tried setting group means to zero in the Mundlak models we found the AUC exactly corresponded to the standard GLMs. This was at first surprising as the coefficients at the individual level are different (since the Mundlak estimates are influenced by the group-mean terms being also present at model fit), however it is a consequence of how the AUC is calculated wherein the threshold is varied to go from a false-positive rate of zero to 1. Since the standard GLM and Mundlak GLM involve the same linear combination, with different weights, the ROC is identical.

Figures:

1. Figure 1. This isn't a proper flow chart. Branches should be disjoint options as the top three boxes in the middle (prevalent CAD, incident CAD, no CAD), but the 4th and 5th boxes and the box on the right (complete covariates) are sequential. Make [Total (left box)] -> [prevalent, incident, no CAD (3 boxes)] -> incident and no-cad flow forward and split on Y/N have PRS -> yes PRS move forward on Y/N PCE score -> Yes PCE move forward on Y/N other covariates -> Y move to study as usable incident and no-CAD participants. We recreated the flowchart to show the process of getting the complete dataset. This is the new Figure 2 in the paper and for convenience is provided as Figure 3 below.
2. Figure 2. Distracting that 2a is a different format than the (b) and(c). Maybe move (a) to its own figure earlier in the paper when it's mentioned that the focus of the paper is on relative mean prediction performance in various substrata of the UKB data set, and the particular groups of interest will be the recruitment centres.
We have separated the original three maps into two separate figures. Thank your for this suggestion.
3. Figure 3. In the caption, mention that the dotted lines are for group means (medians?) They are the means. We have now noted this in the caption.

Figure 3: Flowchart of the complete dataset generation.

4. Figure 8. Some plotting error in panel (b) a lot of lines are missing. With this choice of y-axis range, predicted rates for Barts would be off-scale This version of the plot is no longer in the paper.
5. Figures 9,10,17. These are correlations, not p-values. Fix labels on color scale and captions. Thanks for spotting this error. We have fixed the legend and captions.
6. Figures 9, 10. Except for PCE, everything in Figure 9 is also in Figure 10. Combine these. Done.
7. Figure 10. Note the sign flips in (a) vs (b) for $\text{cor}(\text{PRS}, \text{HDL})$ and $\text{cor}(\text{systolic blood pressure}, \text{TDI})$ by outlining those boxes in the plots, and in the caption. We have noted this and highlighted in bold directly on the figures, as suggested.
8. Figure 20. Why highlight the model with PRS alone? Figure 12 (the other prediction interval figure) highlights PRS + PCE, and by Table 10 PRS+PCE has better performance for the age stratified data. Probably should highlight PRS+PCE results. We compare results from PRS, PCE, PRS+PCE, and one best model from the remaining models. This is now also noted in the paper.

Tables:

1. Table 1. The row and column labels are "AUC CAD definition". Should be "PRS model CAD definition"

Figure 4: The Pearson correlation for (a) Correlation for the complete data set and (b) Correlation using mean values from UKB assessment centres. Correlation coefficients are added in the boxes for pairs of variables where the signs of the coefficients are opposite between (a) and (b).

As suggested by the third Reviewer, we do not actually use LDpred2 and have therefore removed this unnecessary table.

- Table 6. AIC, BIC, LL, Deviance aren't informative unless you're comparing models or model fits, drop these.

Dropped as suggested.

In model specification (which variables used), generally "PCE" is used (say in Table 5). Why is it "PCE risk" in Table 6? We have amended original Table 6 (Table 2 in the revised version) to simply PCE.

Are the variables scaled in a way for the regression coefficients to have interpretable units? What are the units?

No, we did not scale the variables as our goal is not to interpret the model coefficients.

- Table 10. For more consistency with other tables, update column names to ("Variables", "GLM AUC" and "Mundlak GLM AUC") Done. We have also merged the tables.

4 Comments from the third reviewer

This study investigates how coronary artery disease (CAD) can be predicted from genetic and non-genetic variables obtained for UK Biobank (UKB) participants. The main focus is at predicting risk at group level i.e. predicting the fraction of individuals in a particular assessment center that will get a diagnosis. The authors find that in making such predictions it appears beneficial to include group mean effect (Mundlak model). They find that the polygenic risk score (PRS) seems to have a groups level effect.

Strength and weaknesses:

The study is based on a very large cohort from the UKB with genetic and clinical risk estimated available more than 250,000 individual with more than ten years of follow-up. Obtaining accurate estimates of risk of CAD is of obvious clinical relevance as pharmacological prevention is readily available, and important questions in the area is how PRS interplay with non-genetic risk factors and how ascertainment effects and geographical biases may impact on such risk estimates. However, the specific aim of this study is "predicting the risk of CAD at the regional/country level", meaning the authors are interested in predicting the fraction of individuals in a UKB assessment center that will develop CAD, more than predicting who will develop CAD, which I find a bit obscure. Further, many on the central points highlighted as main finding could in my reading only be trends in the data as all estimates lack confidence intervals, and no statistical tests are applied to quantify the statistical significance of the findings.

We have added confidence intervals to the plots showing predicted and observed rates. We have also expanded upon the "Statistical Tests" section to include several more tests. More importantly, we agree that our study did not originally present a sufficiently clear motivation. We therefore expanded the Study Aims section, citing a new paper on predicting rates across countries for a range of diseases, based on genetic samples: [Jain et al., 2023] report 4/14 common disorders having statistically significant correlation between observed prevalence and PRS of a sample within Europe and 8/14 across a global set of countries. Most interestingly, they don't appear to correct for multiple comparisons and the reported correlations (Supplementary Tables 7 and 8) are very low for the statistically significant disorders. We believe our Mundlak model approach leads to a substantial improvement on these results.

Overall assessments While I feel that the general topic is of broad interest, this specific study takes a somewhat unusual angle on the topic, and no convincing explanation why this approach is useful is provided (See Major Comments 9a-b). The general lack of confidence intervals, p-values or other means of justification that the trends described are anything but sampling variance unfortunately means, that I do not find that the results presented provide strong support for the conclusions reached.

Thank you for this constructive criticism. We agree that we did not provide sufficient validation and motivation in the original version. In our revised paper we have addressed this and we provide a point-by-point response to your concerns below.

Major Comments:

1. I find it quite hard to follow the study flow chart in Figure 1. Some boxes seem to contain individuals that were excluded while other seem to be different subset of individuals that were included, but it's unclear for me how to read the diagram. My understanding from the text is that the 263,087 individuals were selected based on: (a) being in the UKB, (b) being classified as white British, (c) not having CAD before joining UKB, (d) having an available CAD-PRS, (e) not having an ASCVD event before joining UKB, (f) LDL-C_i 190, (g) not having extreme blood pressure or cholesterol, and (h) not having missing values for PCE variables.

I think a more traditional stepwise study flow diagram could improve the transparency of the study. For instance, one showing how many individuals were excluded by each of the steps (a-h) and how many remained.

We have created a new flow chart as you and the second reviewer suggested. This is Figure 3 in this response document and Figure 2 in the revised paper.

2. The first result of the manuscript is that there is "geographical variations in age-standardized rates of CAD between assessment centres". Here the authors de-

scribe how they calculated a weighted sum of the fraction of individuals with prevalent CAD in each of eight age-strata with weights corresponding to the European standard population distribution.

- (a) I'm struggling to find the actual estimates. I see some colored dots in Figure 2b, but the unit is not given anywhere (are these percentages?).

They are percentages. Thanks for spotting this error, which we have corrected. These are provided in Table 1 below and Table 4 of the revised paper.

Table 1: Observed CAD rates for 22 UKB assessment centers are presented. The Prevalence column records the age-standardized CAD prevalence rates (in percentage) in 2010, while the Incidence column records CAD incidence rates (in percentage) from 2010 to 2021, calculated based on all new CAD events during that period.

Centre	Prevalence (%)	Incidence (%)
Cardiff	1.958	2.9
Bristol	1.993	3.591
Reading	2.105	3.358
Hounslow	2.114	3.285
Edinburgh	2.323	3.804
Croydon	2.378	3.193
Stockport (pilot)	2.465	5.568
Barts	2.539	2.794
Nottingham	2.627	3.824
Swansea	2.743	2.102
Sheffield	2.841	3.712
Birmingham	2.86	4.01
Oxford	2.968	3.95
Newcastle	3.097	4.603
Leeds	3.159	3.998
Bury	3.205	4.857
Middlesbrough	3.303	4.094
Liverpool	3.434	4.294
Manchester	3.445	4.847
Stoke	3.649	4.781
Glasgow	3.929	4.545
Wrexham	7.994	2.844

- (b) Additionally, no confidence intervals are provided, and no statistical test is applied to demonstrate that the age-standardized prevalence and incidence differ more across the assessment centers than would be expected by chance (i.e. due to measurement error). Without such a test, I find it hard to judge whether the claim of "geographical variations in age-standardized rates of CAD between assessment centres" is supported by the data.

Thank you for this. We run a GLM on the overall data set, using CAD status as the explanatory variable and centre as the independent variable. The p-value for the centre variable is 0.00409, which indicate the significant difference of CAD incidence rates between centres. In addition, we also run GLMs on the

Figure 5: Pair-wise GLM between any two centres, p-value after BH correction.

subset of data containing only two different centres from the complete data set. In total, there were 210 pairs of GLMs, of which 117 pairs had BH adjusted p-values less than 0.05 and 93 pairs had adjusted p-values greater than 0.05. We now include Figure 5, which shows the results of the pairwise tests of the CAD incidence rates in assessment centres, from which we observe many statistically significant differences.

- (c) would it be relevant to standardize the prevalence rates by factors other than age (e.g. sex)? The PCE risk for CAD has been shown to have different predictive accuracy in older and younger people (e.g. [Riveros-Mckay et al., 2021]). We are interested in predicting the CAD risk for a new group, a group that is

most commonly defined by region, or age. So we did not standardise the rates by other factors than age.

3. A second result is the that both the PGS and PCE seem differs between assessment centers.

(a) While we do get many technical details about the calculation of the LDpred2 PRS (which wasn't used for the main analyses anyway) we get very few for the UKB-CAD-PRS. I'm guessing that the PRS being used is this one (<https://biobank.ndph.ox.ac.uk/uk>) and while I can find some information in Supplementary Tables of Thompson et al, I think this information should be summarized in the methods section (e.g. what was the training sample of PGS, what was the sample size, phenotype definition etc.)

Thanks for pointing this out. We began with LDpred2 (hence the detailed description), but switched to the provided resource once we confirmed its superiority. We have now removed details about LDpred2 from our manuscript and added the suggested details on UKB resources category 300 (UKB-CAD-PRS).

(b) Should there be some multiple testing correction given the large number of tests (22x22) presented in Figures 4, 5 and 6? I agree that there seems to more significant ($p < 0.05$) differences than we would expect by chance, but I think a formal test would be appropriate.

Following your suggestion, we performed BH corrected permutation tests. See also our response to the 2nd comment from the first reviewer.

4. A third result is that the authors try to predict CAD using a set of risk factors including PGS, PCE, BMI and TDI. AUC values for are provided in Table 4, but without confidence intervals. In particular, I find that the correlation coefficients also provided in Table 4 are hard to interpret without a confidence interval since if I understand it correctly, the are based on only 22 observations. We have amalgamated all AUC and correlation results into a single table (new Table 2) and included confidence intervals in all cases. We used the DeLong method (see our revised manuscript for details) for the AUC confidence intervals. You are correct that the LOCOCV correlations are based upon only 21 centres (the pilot centre Stockport is not included) and as such the confidence intervals are quite wide.

5. With reference to Figure 7, the authors describe the correlation between observed and predicted case rates at the different assessment centers as low. To show that this correlation is lower than expected, the authors use a rather untraditional approach in which they create groups with different rates of cases and controls. They show that in this random grouping, the correlation between the observed case rates and predicted case rates appears higher. However, I think this analysis should be extended in several ways:

Thanks for this comment. This result demonstrates the ability of GLM to distinguish between low and high incidence groups for the data set, which was created under the designed labelling method. We're not trying to show (using the permutations / relabelling) that the correlation is lower than expected under some hypothesis, but rather we simply observe that it is low and that we'd like it to be higher. We can simply say that the variance explained (as measured by r^2) is low and that our proposed Mundlak model is better. It's a question of model fit, not of significant difference from

a null hypothesis. Therefore we didn't extend this part in our revised version, but we did extend the simulation study related to the original Figure 16, based on your comment.

- (a) How many random groups were there? although it says nine groups the text, I see 15 dots in Figures 8a. Why not 21 as in the actual data? There were originally 15 groups, however we have added more simulation studies in the revised version, which has 21 groups to match the real number of centres. This has necessitated a change of the weights α to $(-0.5, 0.5)$ to recreate groups with a similar spread of group-rates as the assessment centres.
- (b) Why were similarly sized groups picked here? It looks like the outlying groups in the real data tend to be the smaller assessment centers, which we would also expect to have noisier estimates, right? Thanks for this suggestion, which we agree brings the permuted data closer to the real-world data. We now also simulate groups with the sample sizes as the original assessment centres, as well as rates that are highly correlated with the centres' rates. See the revised text on the structured permutation study.
- (c) Was any statistical test used to quantify the difference between Figures 7 and Figure 8a, or just visual inspection? Only visual inspection in the first version. The refocus on the simulation using alpha weights (rather than case-control swaps) allows for a clearer demonstration of the impact of group-means on these correlations.
- (d) Was the random grouping just done once? Why not do it 10,000 times to get an empirical distribution of the correlation coefficient under the null? I think all of these questions above could be addressed by a more traditional permutation test, where samples are permuted within cases and within controls.

In the revised paper, an extended version of the simulation study is designed to better inform the scenarios where the Mundlak model has the ability to reveal the group level effect of PRS and can be used for the group level risk prediction; apologies for the lack of clarity. With regard to the suggestion of a more traditional permutation test wherein samples are permuted within cases and within controls, this would each time yield draws from a null distribution corresponding to no statistically significant difference between rates across assessment centres, but this is not what we wish to test against. We are not concerned with establishing that there is a statistically significant difference across centres, as that is established in the section "Statistical tests" and previously in the cited literature. Rather we wish to demonstrate that despite this variation, a GLM fit to PRS and other covariates fails to model much of the inter-centre variation. We then demonstrate how the Mundlak model specification addresses the issue, leading to more accurate group-rate predictions. The text in the section "Results for UKB assessment centres" has been amended to explain this.

- 6. In Figures 9-10 the author's again make a comparison between correlation coefficients estimated based on $> 280,000$ data points and 21 datapoints given that the correlation coefficient has a standard error $=\text{sqrt}(1 - r^2)/\text{sqrt}(n - 2)$, I'm unsure which of the group-level correlation coefficients should be considered significant. I have a hard time following interpretation of these results. Figures 9-10 gave P-values from Pearson's correlation tests for the complete data and for the group means respectively, they did not provide confidence intervals. Simpson's paradox describes

the situation where a trend appears in several groups of data, but this trend disappears or is reversed when all of these groups are combined. These results are now combined in new Figure 7 and we have added text on the statistical significance of the pairwise correlations at both the level of the individual and at the level of the assessment centre.

7. The section at page 10 lines 43-60, a seems a bit speculative and without a simulation experiment or a DAG analyses to back it up, I think it shouldn't be included as a result in this paper.

We have added a paragraph to state that this is only one possible mechanism through which a reversal of correlation from individual to group level might occur. We do not claim it is the primary mechanism that causes the phenomenon, but our approach corrects for it regardless. The simulation described in the section "How the Mundlak Model Works" generates such a structure in the data.

8. The main result of the paper is to fit a Mundlak model, where in addition to the individual PRS and PCE, two new variables (group mean PRS and group mean PCE) are created which contain the per assessment-center average value of the PRS and PCE, respectively. In Table 6 we see that there seems to be a significant effect of group mean PRS but not of group mean PCE. However, comparing the AUC in Table 4, i.e. without the Mundlak effect (AUC-PRS=0.6318) to that in Table 5, i.e. with the Mundlak effect (AUC-PRS=0.635) the effect seems to account for rather little variability in CAD risk. Looking closer at Table 6, it seems that the beta-value for the group mean PRS is much higher than that for the PRS. My intuition is that this difference indicates that the PRS is somehow capturing other effects that causal genetic effects. Do you agree?

Yes, that is also our understanding. The Mundlak model has the ability to capture the effect of a hidden variable that is not included in this model. Interestingly, this is confirmed by [Abdellaoui et al., 2022] who we now reference and who use a similar model wherein they include a regional correction term for PRS. They do not attempt group rate prediction (our primary goal) but do characterise the increased predictive performance of such models (at the level of the individual) as being a form of indirect accounting for hidden gene-environment correlations. We have expanded on this idea with reference to and discussion of that paper.

9. The authors state that at the correlation as the assessment center level is higher for the Mundlak model than the simple GLM.

- (a) First of all, I am not really able to think of a situation in which you would want to make a prediction at the assessment center level. Statins, which the authors mention a motivation for predicting CAD, are usually prescribed at individual level, not at assessment center level. If the authors have a particular use of such predictions in mind, I would make it more explicit.

We agree that this wasn't originally made sufficiently clear. We have added additional text to the Study Aim section, highlighting why this is of interest. A paper examining this very topic has been published since our paper was reviewed ([Jain et al., 2023]) and we now cite this and comment on the relevance for our study.

Also, [Eletti et al., 2022] describe quantifying disparities in cancer survival between different subgroups (such as those defined by geography) as one of the

primary aims of population cancer epidemiology. They mention that cancer survival is typically used as a proxy for the overall effectiveness of the healthcare system in the treatment and management of cancer. They state that “Cancer research strives to provide an accurate picture of the evolving cancer burden, as well as documenting existing inequalities, using a variety of key indicators, including cancer survival. ” We have added this reference as another example of motivation for our study and approach to our revised paper.

(b) In the last sentence of the Conclusions paragraph “screening programmes and early interventions” are mentioned. If the authors are aware of any initiatives to implement such things at UKB assessment center level, and if that was the real motivation for running the study, I would suggest pointing it out early in the introduction. [Eletti et al., 2022] formulate cancer control strategies, prioritize cancer control measures, and evaluate the effectiveness of national cancer plans after they have been implemented by assessing their impact on survival, modeling spatial effects. We have added discussion of this to the “Study Aim” section.

(c) I agree with the authors that the cross-validated model is a good choice when evaluating the correlation as the assessment center level. Bearing in mind that you have only 22 assessment centers and that you are fitting models in which the explanatory variables are constructed be clustered at assessment center level overfitting seems likely. Indeed, all correlations presented in Table 5 and Table 6 are higher than the cross-validated Mundlak model version presented in Table 8 and Table 9. If the authors agree that the correlations in Table 5 and Table 6 and the corresponding figures (Figure 11, 12 and 13) are overfitted is there any reason to include them in the paper?

We agreed with this comment and have removed the in-sample results, allowing us to place a greater focus on the cross-validated results. This has also allowed for a greater consistency and simplification of figures and we thank the reviewer for the suggestion.

(d) Given that Table 5 and Table 6 are likely overfit Tables 8 and 9 seem to be the right place to judge if the Mundlak model is effective. However, without confidence intervals of the correlation coefficients it is difficult to judge whether the estimated correlations are significantly higher. We have added confidence intervals to all LOCOCV figures. Our new Table 2, which consolidates previous tables to show both AUC and LOCOCV correlation, now also reports confidence intervals.

Minor comments

1. Figure two why does the legend have same label in b) and c) Thanks for pointing this error out. We have corrected it.
2. Figures 9, 10, 17 says “p-values”, but I assume these are correlation coefficients. I would like to see the actual p-values, though. Following the reduction in the number of figures suggested by all three reviewers, we now present a single figure depicting correlation coefficients, replicated here for convenience in Figure 4. With regard to p-values, only TDI and PCE returned a non-significant p-value even after correction for multiple testing using Benjamini & Hochberg corrections at the level of the individuals. Text has been added to the paper to reflect this result as well as the pairs of covariates that are statistically significantly correlated at the group-level.

3. Table 2: "4. Randomly (...) then move the same number of controls from group B to group A." should be "from group D to group C", right? Correct and thank you. This section has been rewritten.
4. Page 4 line 31: "Of the 502,401 UKB participants, 35,308 were identified as participants with a first ASCVD epidemic event." I don't know what epidemic refers to here, but it sounds misplaced. This should have read "episodic". It has been changed.
5. Table S6: "bold italics are the highest in that column" -0.287 seems to be the lowest? That should have read highest in magnitude and has been corrected.

References

- [Abdellaoui et al., 2022] Abdellaoui, A., Dolan, C. V., Verweij, K. J., and Nivard, M. G. (2022). Gene–environment correlations across geographic regions affect genome-wide association studies. *Nature Genetics*, 54(9):1345–1354.
- [Abdellaoui et al., 2019] Abdellaoui, A., Hugh-Jones, D., Yengo, L., Kemper, K. E., Nivard, M. G., Veul, L., Holtz, Y., Zietsch, B. P., Frayling, T. M., Wray, N. R., et al. (2019). Genetic correlates of social stratification in great britain. *Nature Human Behaviour*, 3(12):1332–1342.
- [Camargo et al., 2008] Camargo, A., Azuaje, F., Wang, H., and Zheng, H. (2008). Permutation–based statistical tests for multiple hypotheses. *Source Code for Biology and Medicine*, 3:1–8.
- [Eletti et al., 2022] Eletti, A., Marra, G., Quaresma, M., Radice, R., and Rubio, F. J. (2022). A unifying framework for flexible excess hazard modelling with applications in cancer epidemiology. *Journal of the Royal Statistical Society Series C: Applied Statistics*, 71(4):1044–1062.
- [Hou et al., 2023] Hou, K., Ding, Y., Xu, Z., Wu, Y., Bhattacharya, A., Mester, R., Belbin, G. M., Buyske, S., Conti, D. V., Darst, B. F., et al. (2023). Causal effects on complex traits are similar for common variants across segments of different continental ancestries within admixed individuals. *Nature Genetics*, 55(4):549–558.
- [Hu et al., 2023] Hu, S., Ferreira, L. A., Shi, S., Hellenthal, G., Marchini, J., Lawson, D. J., and Myers, S. R. (2023). Leveraging fine-scale population structure reveals conservation in genetic effect sizes between human populations across a range of human phenotypes. *bioRxiv*, pages 2023–08.
- [Jain et al., 2023] Jain, P. R., Burch, M., Martinez, M., Mir, P., Fichna, J. P., Zekanowski, C., Rizzo, R., Tümer, Z., Barta, C., Yannaki, E., et al. (2023). Can polygenic risk scores help explain disease prevalence differences around the world? A worldwide investigation. *BMC Genomic Data*, 24(1):70.
- [Lin et al., 2023] Lin, B. D., Pries, L.-K., van Os, J., Luykx, J. J., Rutten, B. P., and Guloksuz, S. (2023). Adjusting for population stratification in polygenic risk score analyses: a guide for model specifications in the UK Biobank. *Journal of Human Genetics*, pages 1–4.

[Riveros-Mckay et al., 2021] Riveros-Mckay, F., Weale, M. E., Moore, R., Selzam, S., Krapohl, E., Sivley, R. M., Tarran, W. A., Sørensen, P., Lachapelle, A. S., Griffiths, J. A., et al. (2021). Integrated polygenic tool substantially enhances coronary artery disease prediction. *Circulation: Genomic and Precision Medicine*, 14(2):e003304.

[Yang et al., 2021] Yang, C., Starnecker, F., Pang, S., Chen, Z., Guldener, U., Li, L., Heinig, M., and Schunkert, H. (2021). Polygenic risk for coronary artery disease in the Scottish and English population. *BMC Cardiovascular Disorders*, 21(1):1–9.

May 7, 2024

RE: GENETICS-2024-306965

Dr. Michael Salter-Townshend
University College Dublin
School of Mathematics and Statistics
Science Centre East
Dublin, N/A
Ireland

Dear Dr. Salter-Townshend:

Congratulations! We are delighted to inform you that your manuscript entitled "How group structure impacts the numbers at risk for coronary artery disease: polygenic risk scores and non-genetic risk factors in the UK Biobank cohort" is acceptable for publication in GENETICS. Many thanks for submitting your research to the journal.

The reviewers had a few suggestions for improving the manuscript that you may want to consider. You can view their comments at the bottom of this email.

To Proceed to Production:

1. Format your article according to GENETICS style, as discussed at <https://academic.oup.com/genetics/pages/general-instructions>, and upload your final files at <https://genetics.msubmit.net>.
2. Your manuscript will be published as-is (unedited-as submitted, reviewed, and accepted) at the GENETICS website as an Advanced Access article and deposited into PubMed shortly after receipt of source files and the completed license to publish. Please notify sourcefiles@thegsajournals.org if you do not wish to publish your article via Advanced Access.
3. We invite you to submit an original color figure related to your paper for consideration as cover art. Please email your submission to the editorial office or upload it with your final files. You can submit a small-sized image for evaluation, and if selected, the final image must be a TIFF file 2513px wide by 3263px high (8.375 by 10.875 inches; resolution of 600ppi). Please avoid graphs and small type.

If you have any questions or encounter any problems while uploading your accepted manuscript files, please email the editorial office at sourcefiles@thegsajournals.org.

Sincerely,

Noah Zaitlen
Associate Editor
GENETICS

Approved by:
Hongyu Zhao
Senior Editor
GENETICS

note: Please add jnls.author.support@oup.com and genetics.oup@kwglobal.com (or the domains @oup.com and @kwglobal.com) to your email program's "safe senders" list. You will be contacted by both at various points during the production process.

Review comments (if applicable):

Reviewer #2 (Comments for the Authors (Required)):

Minor comments and requests for clarification:

Page 4, line 93

"35,308 were identified as participants with a first ASCVD episodic event. An additional 155 participants did not have a corresponding date of first ASCVD episodic event, but we still included them in the first ASCVD event group. In total, there are 35,887 UKB participants with first-ever ASCVD. "

Please clarify who these numbers relate? 35887 is not 35308+155. It may be clearer to say the 35887 had "prior ASCVD events" if the point is that they were excluded because their CVD event were prior to joining the UKB.

Page 5, figure 2

"Remove participants who were identified with first-ever CAD before joining the UKB [and] died before September 2021"

should it be "or" rather than "and"?

Page 5, figure 2

"Remove participants who were not eligible for PCE scores calculation:

... "LDL-C < 190 mg/dL"

The other items on the list are exclusion criteria. I think the exclusion criterion for LDL-C is "LDL-C \geq 190 mg/dL"

Page 7, line 24

"When the weights α_1 and α_2 have opposite signs, the relation ship among groups is not as straightforward. For example, if we set $\alpha_1 = -0.5$ and $\alpha_2 = -0.5$,"

For this example, α_2 should be positive based on the rest of the text in the paragraph

Page 8, line 6

comparing this paragraph to Figure 1, should the total number of participants be 502,401 or 502,410?

On page 8 "19,047 participants had their first CAD event after they joined the UKB" but from page 4 and figure 2, 8458 incident CAD events are studied. That difference must be due to the other inclusion/exclusion criteria for analysis as shown in Figure 2, but may be worth mentioning again that the 8458 incident cases were the ones analyzed.

Also, the study flow chart exludes less than half of the cohort (502,401 -> 263,087) but over half of the incident cases (19047 -> 8458). In addition to the inter-site heterogeneity that is the focus of the study, this suggests an incidence rate as a percentage in the UKB participants who were not included in this analysis of 4.4 -- around the highest site incidece rate based on figure 3.

Overall incidence as fraction

> 19047/502401

[1] 0.03791195

Incidence in study

> 8458/263087

[1] 0.03214906

Incidence in UKB participants not included in study (prevalent cases, but also non-white British...)

> (19047-8458)/(502401-263087)

[1] 0.04424731

Reviewer #3 (Comments for the Authors (Required)):

All the questions I had regarding the statistical analyses and interpretation of results have been answered.

While I was not originally very convinced about the relevance of making predictions at assessment center level, the authors have now provided a motivation for this, which seems reasonable.

I have no further comments.